# Semantics and Spatiality of Emergent Communication

**Rotem Ben Zion**[1]    **Boaz Carmeli**[1]    **Orr Paradise**[2]    **Yonatan Belinkov**[1]

[1] Technion – Israel Institute of Technology    [2] UC Berkeley

`rotm@campus.technion.ac.il`    `boaz.carmeli@campus.technion.ac.il`
`orrp@eecs.berkeley.edu`    `belinkov@technion.ac.il`

## Abstract

When artificial agents are jointly trained to perform collaborative tasks using a communication channel, they develop opaque goal-oriented communication protocols. Good task performance is often considered sufficient evidence that meaningful communication is taking place, but existing empirical results show that communication strategies induced by common objectives can be counterintuitive whilst solving the task nearly perfectly. In this work, we identify a goal-agnostic prerequisite to meaningful communication, which we term *semantic consistency*, based on the idea that messages should have similar meanings across instances. We provide a formal definition for this idea, and use it to compare the two most common objectives in the field of emergent communication: discrimination and reconstruction. We prove, under mild assumptions, that semantically inconsistent communication protocols can be optimal solutions to the discrimination task, but not to reconstruction. We further show that the reconstruction objective encourages a stricter property, *spatial meaningfulness*, which also accounts for the distance between messages. Experiments with emergent communication games validate our theoretical results. These findings demonstrate an inherent advantage of distance-based communication goals, and contextualize previous empirical discoveries.

## 1   Introduction

Humans use language in multi-agent social interactions. Pressures of synchronization and collaboration play a central role in shaping the way we communicate. Motivated by this observation and the goal of creating artificial agents capable of meaningful communication, the field of emergent communication (EC) employs a multi-agent environment jointly trained to accomplish a task that requires active transmission of information. The agents utilize a messaging channel that usually takes the form of a discrete sequence of abstract symbols, resembling the structure of human language. Successful optimization results in synchronized agents operating a newly developed communication protocol tailored to the objective. We study a type of EC setup inspired by Lewis games [32] where a sender agent describes a given input and a receiver agent makes a prediction based on that description. The game objective is designed to make the receiver demonstrate knowledge of the original input, which in turn compels the sender to encode relevant information in the message. There are two common objectives used in this type of EC setup (illustrated in Figure 1):

**Reconstruction**  In the reconstruction task [34, 44, 45], the original input is re-generated based solely on the message. We are specifically interested in a reconstruction objective that quantifies distance between prediction and target, forming a discrete version of autoencoding [18].

**Discrimination**  In the discrimination task [17, 28, 33], the original input is retrieved from a set of candidates, incentivized by negative log-likelihood.

38th Conference on Neural Information Processing Systems (NeurIPS 2024).

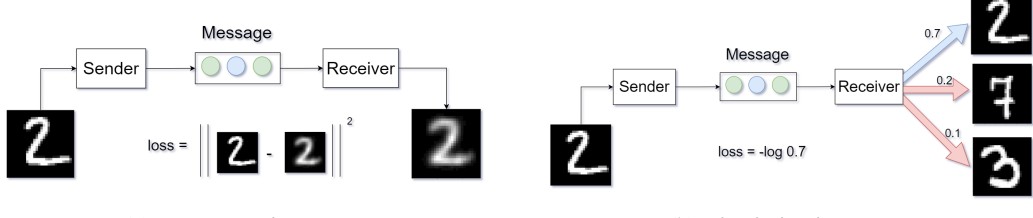

(a) Reconstruction game.

(b) Discrimination game.

Figure 1: Illustration of the reconstruction and discrimination tasks. The discrimination receiver is given the candidates in addition to the message.

A central goal in EC, which motivates our work, is understanding how different factors in the environment affect the emergent protocol, and specifically developing agents and objectives that create protocols with similar characteristics to natural language [2, 25, 40, 41, 45]. Tools and experiments have been developed to evaluate the proximity to natural language by testing for properties such as compositionality [6] or efficiency [7]. Unfortunately, many of these empirical methods show great dissimilarity to human communication. One particularly surprising experiment [3] revealed that protocols created via the discrimination task can generalize extremely well to random noise data, suggesting that they do not signal human-recognizable (meaningful) properties of the input.

In this paper, we identify a fundamental property of any meaningful communication protocol, and thus a prerequisite for similarity to natural language. We observe that the discrete bottleneck forces EC protocols to be many-to-one mappings, i.e., that messages likely represent more than one input. With this in mind, we claim that inputs mapped to the same message should be *semantically similar*, as is the case with human language. We formalize this idea with a mathematical definition that we term **semantic consistency**. We further develop a stricter version of this definition, **spatial meaningfulness**, which also accounts for distances between messages, and is therefore better suited to indicate similarity to natural language.

Armed with these definitions, we analyze theoretical solutions to the two common EC environments. In the reconstruction setting, under mild assumptions, we prove that **every optimal solution is semantically consistent**. With a different set of assumptions, we also show that reconstruction-induced communication protocols are spatially meaningful. In sharp contrast, we surprisingly find that the discrimination objective **does not guarantee semantic consistency**, i.e., a communication protocol can be optimal in a discrimination environment but still not semantically consistent nor spatially meaningful. In fact, we show that uniformly random messages can lead to a globally optimal discrimination objective value, meaning that the relationship between inputs mapped to the same message is potentially arbitrary despite optimally solving the task. Our results provide theoretical support to previous empirical findings, such as the discrimination generalization to noise. We further analyze several common variations of the discrimination game from the EC literature.

To support our findings, we run experiments on Shapes [27] and MNIST [30]. The empirical results agree with most of our theoretical findings. However, we observe a gap between theory and practice regarding the level of channel utilization, which we leave for future work to further investigate.

## 2 Background and related work

### 2.1 Emergent communication with Lewis games

A large portion of EC research, based on Lewis signaling games [28, 32], defines two-agent cooperative tasks where one agent, $S_\theta$ ("sender"), receives an input $x \in \mathcal{X}$ ("target") and generates a message $m \in M$, and the second agent, $R_\varphi$ ("receiver"), takes that message and outputs a prediction. The two most common such tasks are reconstruction and discrimination.

In the reconstruction task, the receiver tries to predict the target directly from the message. This is a generative method, so it can naturally be modeled with a receiver that outputs a distribution over the input space [7, 41, 42, 44], i.e., $R_\varphi : M \to \Delta(\mathcal{X})$. In this case, the objective is usually negative log-likelihood or accuracy. However, this approach has two major flaws: First, explicitly outputting a distribution is not always feasible, e.g., over images. Thus, this game is mostly implemented on simple synthetic datasets. Second, the incentive of likelihood or accuracy does not measure *how*

*close* the receiver's output is to $x$.[1] In this work, we assume a different approach to reconstruction, inspired by autoencoders [18, 34], where the receiver outputs an object in the input space rather than a distribution, i.e., $R_\varphi : M \to \mathcal{X}$. Thus, the loss function in our analysis is straightforwardly the distance between prediction and target.

In the discrimination task, the receiver tries to identify the target within a set of candidates [17, 28, 29, 33, 44]. It can be modeled as $R_\varphi : M \times \mathcal{X}^d \to \Delta\{1, \ldots, d\}$. We explore some alternative formulations in Appendix F. This setting introduces two major choices: the number of distractors (candidates other than $x$) and how to select them [12, 29]. These parameters have been shown to have a major effect on the resulting protocols [15, 29]. The results in the main body of this paper concern the vanilla version of the discrimination task, where candidates are chosen independently. We analyze other discrimination versions in Appendix A.

## 2.2 Towards interpretable human-like communication

A central line of study aims to develop EC systems that create protocols with high proximity to natural language. Some have tried aligning the two explicitly, by using pretrained language models [31, 43] or training translation systems [45, 46]. In this work, we focus on properties of the EC protocol itself, as induced by the objective rather than exposure to text. The most studied such characteristic is compositionality, which refers to the ability of agents to assemble complex units (e.g., sentences) from simpler units (e.g., words). Compositionality is hard to measure [1, 5, 6, 24, 37], and while some claim that it is correlated with generalization [2, 40, 42], the opposite has also been argued [8, 11, 13]. In any case, emergent languages are often not compositional [25], inefficient [7, 41], and generalize to noise [3], suggesting a mismatch between common EC objectives and the evolution of human language. In this work, we advance the goal of creating human-like communication by interpreting properties of natural language into formal constraints, and analyzing whether the EC objectives create protocols that follow them. A commonly used compositionality measure, topographic similarity (topsim) [5], is related to the definitions presented in this work. It evaluates the correlation between distances in the input space and the corresponding distances in the messages space. Notably, topsim considers the relationship between every pair of inputs, whereas our definitions only consider pairs that correspond to similar messages. The latter follows an intuitive asymmetry: inputs with similar messages are expected to be similar, but inputs with dissimilar messages do not have to be different (for example, the same image can often be described by several distinct human captions).

## 2.3 Input information encoded in the message

The game objective dictates the information that needs to be stored in messages. In this work, we focus on spatial information by considering the geometric relationship between inputs mapped to the same message. To the best of our knowledge, no previous work has theoretically analyzed this idea in EC literature. A related concept is the mutual information (MI) between messages and inputs, which has been studied both empirically [1, 15, 23, 35, 45] and theoretically [19, 21, 42]. MI is strongly related to the infoNCE objective [38] often used in implementations of the discrimination game. MI can be used to measure the complexity of the communication channel [44, 45], and is correlated with task performance [23, 45] and even compositionality [1]. Finally, The mutual information gap [9] can be used to measure disentanglement [8, 13], which is also related to compositionality. However, Guo et al. [15] show that MI is not sufficient to indicate transfer-learning generalization, and connect that to pairwise distances of inputs mapped to the same message, effectively testing for semantic consistency. The first part of this work, combined with their findings, would suggest that languages developed by agents in the reconstruction setting should be more expressive and generalize better to different tasks.

On the theoretical side, a relevant work is Rita et al. [42], where the global discrimination objective is decomposed into two interpretable components, a co-adaptation loss and MI. The latter is equivalent to our Lemma A.1. In this work, we use such equivalent objectives for both the reconstruction and discrimination tasks to investigate the behaviour of optimal solutions, and specifically whether the tasks encourage semantic consistency.

---

[1] We refer to this game as "global discrimination" and analyze it in Appendix A.1. When the input space is finite, it is equivalent to a special case of the discrimination game.

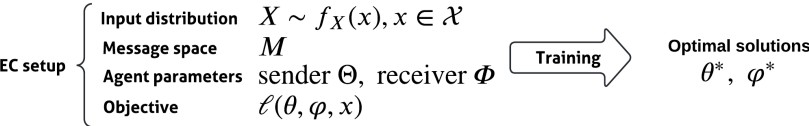

Figure 2: Notation for the emergent communication (EC) setup.

## 3 Notation and task definitions

### 3.1 General notation

We begin by introducing general notation for the EC setup; see Figure 2 for illustration. The input space is modeled by a real-valued random variable $X = (\mathcal{X}, f_X)$, where $\mathcal{X} \subseteq \mathbb{R}^{d_x}$ is the bounded set of possible inputs (e.g., images), and $f_X \colon \mathcal{X} \to \mathbb{R}_+$ is a prior distribution. Denote by $M = \{m_1, m_2, \dots\} \subset \mathbb{R}^{d_M}$ the finite or countably infinite set of possible messages.[2] We assume that the minimal distance between a pair of messages is attained, i.e., that $\min_{m_1 \neq m_2 \in M} \|m_1 - m_2\| \in (0, \infty)$.

The *sender* agent $S_\theta \colon \mathcal{X} \to M$, parameterized by $\theta \in \Theta$, maps each input to a message. Sender $S_\theta$ defines the communication protocol; we compare EC setups based on which optimal sender agents they induce. The *receiver* agent $R_\varphi$, maps the message and optionally additional input to a prediction. It is parameterized by $\varphi \in \Phi$. During EC training, the goal is to learn hypotheses $(\theta, \varphi)$ that minimize a *loss* described by a function $\ell \colon \Theta \times \Phi \times \mathcal{X} \to \mathbb{R}$, which maps a pair of agents and an input to a real value. To group all of the above notation into a single definition, denote an *EC setup* as a sextuple $(\mathcal{X}, f_X, M, \Theta, \Phi, \ell)$. An EC setup induces a set of *optimal* agents by

$$\theta^*, \varphi^* \in \underset{\theta \in \Theta, \varphi \in \Phi}{\arg\min} \; \underset{x \sim X}{\mathbb{E}} \; \ell(\theta, \varphi, x).$$

Note that every part of the setup can affect the set of optimal agents. We also define a notion of conditional optimality, where one of the agents is fixed at a potentially sub-optimal hypothesis; sender $S_\theta$ is *synchronized* with receiver $R_\varphi$ if the pair is optimal for the EC setup $(\mathcal{X}, f_X, M, \Theta, \{\varphi\}, \ell)$. The same definition applies to the receiver agent respectively. Note that a pair of agents can be synchronized in both directions without being optimal.

### 3.2 Task definitions

In the reconstruction setting, the receiver agent predicts the original input (the "target") given only the message. In the discrimination setting, the receiver is given the message and a set of candidate inputs containing the target in a random position, and attempts to predict that position based on the message. A visual illustration can be seen in Figure 1. We now present formal definitions of these setups.

**Reconstruction.** The receiver maps a message $S_\theta(x)$ to a prediction in the input space. The loss is the Euclidean distance between that prediction and the target $x$.

- $\Phi_{\text{reconstruction}}^{M, \mathcal{X}}$ is defined as the set of all functions of the form $R \colon M \to \mathcal{X}$.
- $\ell_{\text{reconstruction}}(\theta, \varphi, x) := \|R_\varphi(S_\theta(x)) - x\|^2$.

**Definition 1.** An EC setup $(\mathcal{X}, f_X, M, \Theta, \Phi, \ell)$ is a *reconstruction game* if $\Phi \subseteq \Phi_{\text{reconstruction}}^{M, \mathcal{X}}$ and $\ell = \ell_{\text{reconstruction}}$. It has an *unrestricted* receiver hypothesis class if $\Phi = \Phi_{\text{reconstruction}}^{M, \mathcal{X}}$.

**Discrimination.** In addition to a message $S_\theta(x)$, the receiver sees a set of candidates $\{x_1, \dots, x_d\}$, which contains the target at a random position $t$, i.e. $x_t = x$, and the rest are $d-1$ independently sampled distractors. The receiver outputs a probability distribution over the candidates. The loss is the negative log-likelihood of the correct position $t$ according to this distribution, averaged over the target position and distractors.

---

[2]The description of $M$ as a vector space is flexible. As an example instantiation, the classic communication channel where each message is a sequence of symbols from a fixed vocabulary can be interpreted as a concatenation of one-hot vectors. In that case, we would have $d_M = V \cdot L$ where $V$ is the vocabulary size and $L$ the message length. The Euclidean distance between messages becomes their Hamming distance.

- $\Phi_{\text{discrimination}}^{M,\mathcal{X},d}$ is defined as the set of all functions of the form $R\colon M \times \mathcal{X}^d \to \Delta\{1,\ldots,d\}$.

- $\ell_{\text{discrimination}}^{X,d}(\theta,\varphi,x) := \displaystyle\mathop{\mathbb{E}}_{\substack{t \sim U\{1,\ldots,d\},\; x_t = x \\ x_1,\ldots,x_{(t-1)},x_{(t+1)},\ldots,x_d \sim X^{d-1}}} - \log P(R_\varphi(S_\theta(x), x_1,\ldots,x_d) = t).$

**Definition 2.** An EC setup $(\mathcal{X}, f_X, M, \Theta, \Phi, \ell)$ is a *d-candidates discrimination game* if $\Phi \subseteq \Phi_{\text{discrimination}}^{M,\mathcal{X},d}$ and $\ell = \ell_{\text{discrimination}}^{X,d}$. It has an *unrestricted* receiver hypothesis class if $\Phi = \Phi_{\text{discrimination}}^{M,\mathcal{X},d}$

In this vanilla version of the discrimination game, the candidates are sampled independently. Several other choice mechanisms have been explored in EC literature. In Appendix A, we define and analyze the most common such variations:

**Global discrimination** In Appendix A.1, we analyze a version of the discrimination game where the receiver outputs a distribution over the entire data rather than a small set of candidates. We find that optimal communication protocols in this setting maximize mutual information between the inputs and messages, as shown in Rita et al. [42].

**Supervised discrimination** Appendix A.2 explores a variant of the discrimination game that incorporates labels by selecting distractors with labels different from the target. We find that optimal communication strategies in this setting are both diverse (the sender is encouraged to output messages uniformly) and pure (labels distribute with low entropy given a message).

**Classification discrimination** In Appendix A.3, we consider a format of the discrimination game where the target is excluded from the candidate set, requiring the receiver to identify a candidate that matches the target's label. We find that optimal solutions in this setup maximize mutual information between messages and labels.

## 4 Semantic consistency: a prerequisite to meaningful communication

In many EC environments, the message is a latent representation of the input, and is often directly analogous to the corresponding latent vector from continuous training setups (e.g. autoencoding, contrastive learning). That said, the discrete bottleneck in EC offers an additional perspective to the concept of a message; when the message space is smaller than the input space, the communication protocol is forced to be a many-to-one mapping, meaning that each message represents a *set* of inputs.

We consider inputs mapped to the same message as *equivalent* with respect to the communication protocol, thus defining a set of equivalence classes that partitions the input space. With this perspective in mind, we wish to understand whether this partition signals meaningful properties of the inputs. In other words, we ask the question: *Is there a meaningful relationship between inputs mapped to the same message?* This motivation is illustrated in Figure 3.

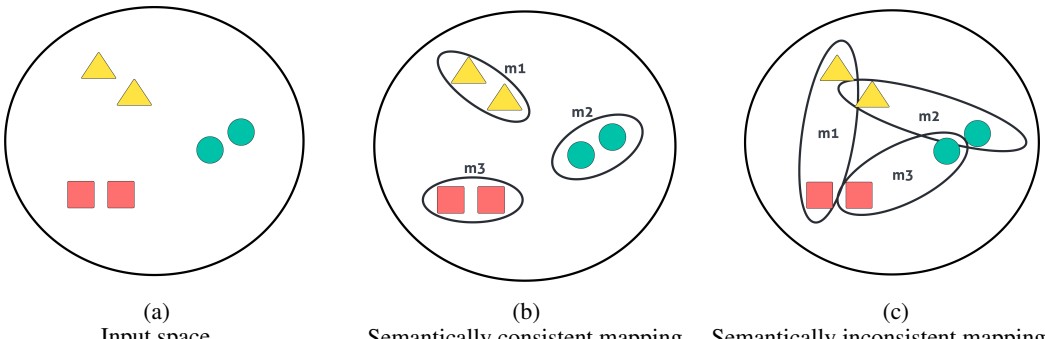

|       (a)        |             (b)              |              (c)               |
| :-------------: | :--------------------------: | :----------------------------: |
|  Input space.   | Semantically consistent mapping. | Semantically inconsistent mapping. |

Figure 3: A message describes a set of inputs. Note: the shapes and colors are not part of the input.

The semantics of a message can be interpreted as the set of properties shared across all inputs in its equivalence class. We would like to design a theoretical test that evaluates whether meaningful properties are encoded in this way in an emergent communication protocol. We term this notion *semantic consistency*, based on the idea that inputs with shared properties are more semantically similar, as illustrated in Figure 3. We use the same idea to develop a formal definition: We define a communication protocol to be semantically consistent if a random pair of inputs mapped to the

same message is, in expectation, more similar than two completely random inputs. Formally, for a communication protocol $S_\theta$, consider the following inequality: [3]

$$\mathop{\mathbb{E}}_{x_1,x_2 \sim X}\left[\|x_1 - x_2\|^2 \ \middle| \ S_\theta(x_1) = S_\theta(x_2)\right] < \mathop{\mathbb{E}}_{x_1,x_2 \sim X}\left[\|x_1 - x_2\|^2\right]. \tag{1}$$

Equation (1) is simplified in Appendix C.1 into the following formal definition:

**Definition 3.** A communication protocol $S_\theta$ is *semantically consistent* if

$$\mathop{\mathbb{E}}_{m \sim S_\theta(X)}\left[\mathrm{Var}\left[X \mid S_\theta(X) = m\right]\right] < \mathrm{Var}\left[X\right].$$

The difference between the two expressions in Definition 3 is the *explained variance*, given by $\mathrm{Var}_{m \sim S_\theta(X)}\left[\mathbb{E}\left[X \mid S_\theta(X) = m\right]\right]$. Thus, a communication protocol is semantically consistent if it explains some of the variance in $X$.

*Remark.* If the message space is larger than the input space, a sender could map each input to a unique message and lose no information. Any such lossless protocol is trivially semantically consistent.

# 5   Are EC systems semantically consistent?

Having defined the two common EC environments in Section 3.2, we now apply Definition 3 to assess whether semantically consistent protocols are induced by reconstruction and discrimination tasks. Formally, we answer the following question for each of the tasks:

*Is every optimal emergent protocol semantically consistent?*

Under the assumption of an unrestricted receiver hypothesis class, we find that:

- In the reconstruction game, the answer is **yes**. We show that equivalent inputs in reconstruction-induced communication protocols are clustered together in input space, matching Figure 3b.

- In the discrimination game, the answer is **no**. The relationship between equivalent inputs in a discrimination-induced protocol can be arbitrary, matching Figure 3c.

All proofs are given in Appendix C.

## 5.1   Reconstruction results

Ideally, we would like to examine a closed-form set of optimal communication protocols. Unfortunately, there is no closed-form solution to the set of optimal agent pairs in a general reconstruction setting. That said, we do have such a solution to the synchronized receiver, i.e., the optimal receiver for a fixed sender $S$ (the following $\arg\min$ set contains a single item):

$$R^*(m) = \mathop{\arg\min}_{r \in \mathcal{X}} \mathop{\mathbb{E}}_{x \sim X}\left[\|r - x\|^2 \ \middle| \ S(x) = m\right] = \mathbb{E}\left[X \mid S(X) = m\right]$$

By assuming that $\Phi$ is unrestricted, we can plug the closed-form synchronized receiver back into the loss function. This yields a transformed objective given in the following lemma.

**Lemma 5.1.** [proof in page 16] *Let* $(\mathcal{X}, f_X, M, \Theta, \Phi, \ell)$ *be a reconstruction game. Assuming* $\Phi$ *is unrestricted, a sender* $S_\theta$ *is optimal if and only if it minimizes the following objective:*

$$\sum_{m \in M} P(S_\theta(X) = m) \cdot Var\left[X \mid S_\theta(X) = m\right] \tag{2}$$

This equivalent objective clearly shows the connection between inputs mapped to the same message: In every optimal solution, these inputs will have high proximity. (In fact, this is the objective function of k-means clustering [36], as elaborated in Appendix B.) Moreover, this formula is the unexplained variance, so minimizing it will maximize the explained variance, leading to the following theorem.

**Theorem 5.2.** [proof in page 17] *Let* $(\mathcal{X}, f_X, M, \Theta, \Phi, \ell)$ *be a reconstruction game. Assuming* $\Phi$ *is unrestricted and* $\Theta$ *contains at least one semantically consistent protocol, every optimal communication protocol is semantically consistent.*

---

[3]We assume that Euclidean distance in the input space entails semantic information. In some types of data, like raw image pixels, using pretrained embeddings may improve the validity of that assumption.

## 5.2 Discrimination results

In a similar fashion to the reconstruction setting, we have a closed form solution to the synchronized discrimination receiver:[4]

$$R^*(m, x_1, \ldots, x_d) = \mathbb{P}\left(t \mid x_1, \ldots, x_d, m\right)$$

Explicitly (see Lemma 5.3's proof):

$$P\left(R^*(m, x_1, \ldots, x_d) = j\right) = \frac{1}{|\{x_1, \ldots, x_d\} \cap S^{-1}(m)|} \cdot \mathbf{1}\{S(x_j) = m\}$$

Applying this receiver, we get the following discrimination loss:[5]

**Lemma 5.3.** [proof in page 17] *Let $(\mathcal{X}, f_X, M, \Theta, \Phi, \ell)$ be a d-candidates discrimination game. Assuming $\Phi$ is unrestricted, a sender $S_\theta$ is optimal if and only if it minimizes the following objective:*

$$\sum_{m \in M} P(S_\theta(X) = m) \cdot \mathbb{E} \, \log\left(1 + \text{Binomial}\big(d - 1, P(S_\theta(X) = m)\big)\right) \tag{3}$$

*And when $d = 2$ (a single-distractor game), this simplifies into: $\sum_{m \in M} P(S_\theta(X) = m)^2$.*

This equivalent objective reveals the interesting disparity between reconstruction and discrimination. Note that this objective is solely a function of the 'sizes' of equivalence classes (the probability for each message). Thus, while the above formula incentivizes the sender to distribute the messages uniformly, it does not impose any constraint on their content; the connection between inputs mapped to the same message could be arbitrary. In formal terms, we prove the following corollary:

**Corollary 5.4.** [proof in page 19] *If the set of possible messages is finite, any sender agent that assigns them such that their distribution is uniform can be paired with some receiver to obtain a globally optimal loss for the discrimination game.*

Using this corollary, we prove that optimal solutions can be inconsistent in the discrimination setting:

**Theorem 5.5.** [proof in page 20] *There exists a discrimination game $(\mathcal{X}, f_X, M, \Theta, \Phi, \ell)$ where $\Phi$ is unrestricted and $\Theta$ contains at least one semantically consistent protocol, in which not all of the optimal communication protocols are semantically consistent.*

## 6 A missing dimension: spatiality in the message space

While the notion of semantic consistency successfully differentiates between the two EC frameworks, it has a major shortcoming: is does not take into account distances between messages. Spatiality in the message space is integral to concepts like compositionality [5] and ease of teaching [33]. A hypothetical communication protocol that maps similar messages to very different meanings may satisfy Definition 3, but is fundamentally different from natural language. We now present a stricter version of semantic consistency, *spatial meaningfulness*, which does consider distances in the message space, and therefore better indicates an inherent similarity to natural language. In addition, we eliminate the assumption that the receiver's hypothesis class is unrestricted, in favor of simplicity and non-degeneracy conditions, which are more realistic in the context of natural language.

To introduce distance between messages, we replace the message equality with similarity in Equation (1). Let $\varepsilon_0$ denote a threshold under which messages are considered similar. To avoid reverting back to Definition 3, we require that $\varepsilon_0$ be greater than or equal to $\varepsilon_M$, defined as the minimal distance between messages: $\varepsilon_M := \min_{m_1 \neq m_2 \in M} \|m_1 - m_2\|$.[6]

**Definition 4.** For $\varepsilon_0 \geq \varepsilon_M$, a communication protocol $S_\theta$ is $\varepsilon_0$-*spatially meaningful* if $\forall 0 < \varepsilon \leq \varepsilon_0$

$$\mathbb{E}_{x_1, x_2 \sim X}\left[\|x_1 - x_2\|^2 \mid \|S_\theta(x_1) - S_\theta(x_2)\| \leq \varepsilon\right] < \mathbb{E}_{x_1, x_2 \sim X}\left[\|x_1 - x_2\|^2\right]$$

A communication protocol $S_\theta$ is *spatially meaningful* if this definition holds for $\varepsilon_0 = \varepsilon_M$.

---

[4] Negative log-likelihood is a proper scoring rule; see Lemma 5.3's proof.

[5] The expectation is taken with respect to the binomial distribution.

[6] This is well defined due to our assumption in Section 3.

Note that this definition is stricter than semantic consistency. Informally, Definition 4 requires the protocol to be semantically consistent when combining pairs of close messages.

With this new definition, we are able to reach similar results to the original semantic consistency:

- In the reconstruction game, for any receiver that satisfies two conditions, we show that every synchronized sender is spatially meaningful.

- In the discrimination game, given the same conditions, we show that a synchronized sender is not necessarily semantically consistent (and hence nor spatially meaningful).

We now define two conditions on the receiver.

**Simplicity constraint.** The first condition limits the receiver's complexity, and introduces the effect of distance between messages. We do that by bounding the rate of change in receiver's output, similarly to a Lipschitz constant. The bounding term depends on the variance of the input distribution and the chosen distance threshold. Formally:

**Definition 5.** Receiver $R_\varphi$ is $(X, M, \varepsilon_0)$-*simple* if $\forall x_1, x_2 \in \text{domain}(R_\varphi)$:

$$\|R_\varphi(x_1) - R_\varphi(x_2)\| \leq k \cdot \|x_1 - x_2\|, \qquad \text{where} \quad k = \frac{\sqrt{2} - 1}{2\varepsilon_0} \sqrt{\text{Var}[X]}$$

**Non-degeneracy constraint.** The second condition prevents the receiver from being too weak. After applying the first condition, we no longer have a closed-form solution for the synchronized receiver. Instead, we take a non-optimal receiver, but add a condition to ensure it is strictly better than any constant function. Formally, let $\varphi^C$ be a receiver that always outputs $C$, and let $C^* = \arg\min_{x \sim X} \mathbb{E}\, \ell(\theta, \varphi^C, x)$ be the optimal such constant. (Note the sender does not matter here.)

**Definition 6.** Receiver $R_\varphi$ is *non-degenerate* if

$$\sup_{x \in \mathcal{X}} \ell(\theta^*, \varphi, x) \leq \frac{1}{4} \cdot \mathbb{E}_{x \sim X} \ell(\theta, \varphi^{C^*}, x)$$

for every sender $\theta^*$ that is synchronized with $R_\varphi$.

## 6.1 Results

With these definitions, we have the following theorems:

**Theorem 6.1.** [proof in page 21] *Let $(\mathcal{X}, f_X, M, \Theta, \Phi, \ell)$ be a reconstruction game, let $\varepsilon_0 \geq \varepsilon_M$ and let $\varphi \in \Phi$ such that $R_\varphi$ is $(X, M, \varepsilon_0)$-simple and non-degenerate. Every sender that is synchronized with $R_\varphi$ is $\varepsilon_0$-spatially meaningful.*

**Theorem 6.2.** [proof in page 22] *There exists a discrimination game $(\mathcal{X}, f_X, M, \Theta, \Phi, \ell)$, $\varepsilon_0 \geq \varepsilon_M$ and a receiver $\varphi \in \Phi$ which is $(X, M, \varepsilon_0)$-simple and non-degenerate, where a synchronized sender matching $R_\varphi$ is not $\varepsilon$-spatially meaningful for any $\varepsilon$.*

# 7 Experiments

To support our theoretical findings, we run experiments on two datasets: (i) the MNIST dataset [30] contains images of a single hand-written digit; (ii) the Shapes dataset [27] contains images of an object with random shape, color and position. We train reconstruction and discrimination (40 distractors) agents using a communication channel of vocabulary size 10 and message length 4 (no EOS token), optimized with Gumbel-Softmax [20, 22]. In the MNIST experiments, we use the digit labels to train another set of agents with the supervised discrimination task (40 distractors). Further data and training details are found in Appendix D.1.

## 7.1 Semantic consistency, compositionality and task performance

To evaluate semantic consistency empirically, we develop the *empirical message variance* measure (in Appendix D.2) which calculates the empirical variance of each equivalence class and takes a

---

[2]Our code is available at `https://github.com/Rotem-BZ/SemanticConsistency`.

weighted average. We measure task performance using discrimination accuracy over 41 candidates (40 distractors), where the referential prediction of reconstruction-trained agents is defined as the closest candidate to the reconstructed target. We also report Topographic Similarity [5], the most common compositionality measure from literature. See Appendix E.1 for results with other compositionality measures (PosDis [8], BosDis [8] and Speaker-PosDis [13]). Additional information on each metric is given in Appendix D.2.

Table 1 and Table 2 show results over the Shapes and MNIST validation sets, respectively (showing standard deviation). While all setups achieve good task performance over MNIST, the discrimination game yields higher discrimination accuracy over the more difficult Shapes dataset, As expected. Interestingly, the discrimination-trained agents utilize less of the communication channel's capacity, as shown by the number of unique messages in both tables. This presents a gap from the theoretical analysis, as the discrimination setting should benefit from maximal message diversity (this can be seen in Lemma 5.3). To enable a fair comparison, we establish individual random baselines that maintain the number of inputs mapped to each message but randomize their assignment. Thus, the baseline's message variance shows the level of message diversity induced by the task, whereas the improvement over the baseline indicates semantic consistency. Over both datasets, the reconstruction task induces significantly more semantically consistent protocols both in absolute value and improvement over the baseline, as predicted by our theoretical findings. Furthermore, TopSim strongly favors the reconstruction setting, suggesting a connection between semantic consistency and compositionality. We note that the difference measured by TopSim is more significant over the Shapes dataset, which is itself more compositional, as objects have several attributes. These results, and the correlations reported in Appendix E.1, indicate little to no correlation between the evaluated compositionality and the discrimination objective performance. In fact, within the set of reconstruction runs on Shapes, their correlation is negative ($-0.24$). This illustrates the counter-intuitive nature of this objective, which we discuss in Section 5.2.

Table 1: Empirical results on Shapes, averaged over five randomly initialized training runs.

| | | | | Msg Var $\downarrow$ | |
| EC setup | Uniqe Msgs | Disc. accuracy $\uparrow$ | TopSim $\uparrow$ | Trained | Rand |
|---|---|---|---|---|---|
| Reconstruction | 306.60 $\pm$28.52 | 31.64 $\pm$2.51 | 0.34 $\pm$0.02 | 1334.38 $\pm$78.05 | 2554.77 $\pm$108.19 |
| Discrimination | 251.60 $\pm$29.53 | 61.96 $\pm$4.78 | 0.09 $\pm$0.01 | 2280.24 $\pm$157.38 | 2793.65 $\pm$115.45 |

Table 2: Empirical results on MNIST, averaged over three randomly initialized training runs.

| | | | | Msg Var $\downarrow$ | |
| EC setup | Uniqe Msgs | Disc. accuracy $\uparrow$ | TopSim $\uparrow$ | Trained | Rand |
|---|---|---|---|---|---|
| Reconstruction | 2523.66 $\pm$30.0 | 88.00 $\pm$0.6 | 0.365 $\pm$0.042 | 371.63 $\pm$2.1 | 1029.57 $\pm$13.9 |
| Discrimination | 402.00 $\pm$70.4 | 78.76 $\pm$10.4 | 0.360 $\pm$0.036 | 1226.14 $\pm$42.4 | 1784.59 $\pm$24.3 |
| Supervised disc. | 287.33 $\pm$28.5 | 87.10 $\pm$5.4 | 0.269 $\pm$0.044 | 1381.72 $\pm$41.6 | 1821.53 $\pm$12.2 |

## 7.2 Message purity

Since the MNIST dataset has only the digit label, we expect meaningful communication protocols to signal digits in their messages. To evaluate this, we calculate the percentage of images that have the majority digit label of their equivalence class. This can be interpreted as the highest attainable accuracy of a classifier mapping messages to digits. Figure 4 shows the most significant improvement over the baseline in the supervised discrimination game, as expected by Lemma A.2. The reconstruction task induces a nearly perfect digit description protocol as well, despite being unsupervised, providing evidence of its ability to unveil meaningful properties, in agreement with our findings. On the other hand, message purity is much lower in the unsupervised discrimination setting, again showcasing that task success does not guarantee an intuitive communication strategy.

In Appendix E.2, we perform a similar analysis over the Shapes dataset, by evaluating message purity with respect to each attribute individually or to a maximum aggregation. We find similar but weaker results.

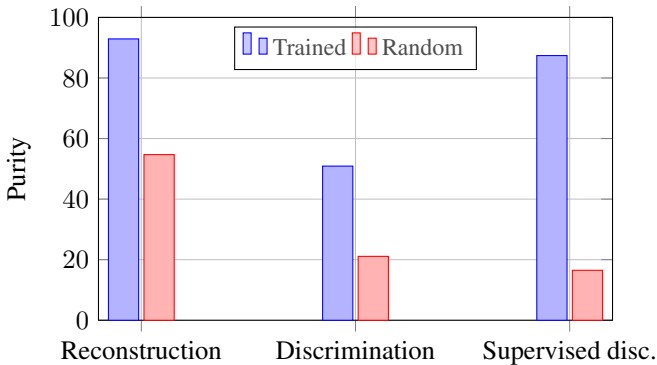

Figure 4: Average message purity, comparing trained models to random baselines.

### 7.3 Spatial meaningfulness analysis

We develop an evaluation method for spatial meaningfulness, which we term *cluster variance*, based on the message variance measure. While we do not find a decisive result using this method, we suggest several ideas for further investigation; see Appendix E.3 for the full analysis.

## 8 Limitations

Our analysis compares the two frameworks via their optimal solutions, and often assumes unrestricted hypothesis classes. In reality, the complexity of the agents is limited by the chosen architectures. Optimization-error affects the emergent protocol as well, especially when it is discrete, as regular differentiable training is not possible. Future work may further analyze these assumptions from the theoretical side, or evaluate their effect empirically. This limitation is evident in our empirical results (Section 7), where channel utilization in the discrimination game behaves differently than expected.

Our main results regarding the discrimination game deal with a simplistic version, where candidates are chosen independently. While we analyze some common variations in the appendix, many other ideas have been proposed and shown to have significant benefits, including multiple-target games [6, 37], different-view candidate representations [10, 16], and multi-modality [14].

Our results assume, as illustrated in Figure 3, that proximity in the input space entails semantic information, and specifically that Euclidean distance broadly indicates the level of semantic similarity. Future work may investigate other distance metrics or evaluate the validity of this assumption.

## 9 Conclusion

Based on properties of natural language, we have defined notions of semantic consistency and spatial meaningfulness, meant to evaluate meaningfulness in emergent communication protocols. Using these definitions, we found that communication protocols generated to solve the reconstruction objective have messages with inherent meaning, while the discrimination objective can be solved by unintuitive systems. Our findings provide insight into known empirical results in EC, such as the ability of agents in the discrimination game to perform well on unseen random noise. The theoretical tools that we have proposed can be used for future investigation and design of EC environments. As a main takeaway, we conclude that distance-based EC environments have promising potential in the prospect of inducing characteristics of natural language, whereas probability-based objectives must be more carefully designed.

## Acknowledgements

This research was supported by grant no. 2022330 from the United States - Israel Binational Science Foundation (BSF), Jerusalem, Israel, the Israel Science Foundation (grant no. 448/20), an Azrieli Foundation Early Career Faculty Fellowship, and an AI Alignment grant from Open Philanthropy. OP was funded by Project CETI via grants from Dalio Philanthropies and Ocean X; Sea Grape Foundation; Virgin Unite and Rosamund Zander/Hansjorg Wyss through The Audacious Project: a collaborative funding initiative housed at TED.

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

# A Analysis of discrimination variations

In this section, we extend our analysis of the discrimination game to three common variations from EC literature.

## A.1 Global discrimination

In this variation, rather than sampling a small set of candidates, the receiver must discriminate between the entire set of data points, which does not have to be finite nor countable. In related work this game is often referred to as the reconstruction game [7, 41, 42, 44], as it can be interpreted as a generative task. However, since it uses the negative log-likelihood objective and not a distance-based one, we call it *global discrimination*. Note that when the set of data inputs $\mathcal{X}$ is finite, this is similar to a special case of the regular discrimination game, where the number of candidates $d$ is set to $|\mathcal{X}|$ (and candidates are sampled without replacement). This equivalence is also mentioned by [12]. Denote:

- $\Phi_{\text{global}}^{M,\mathcal{X}}$ to be the set of all functions of the form $R \colon M \to \Delta(\mathcal{X})$.
- $\ell_{\text{global}}(\theta, \varphi, x) := -\log P(R_\varphi(S_\theta(x)) = x)$

**Definition 7.** An EC setup $(\mathcal{X}, f_X, M, \Theta, \Phi, \ell)$ is a *global discrimination game* if $\Phi \subseteq \Phi_{\text{global}}^{M,\mathcal{X}}$ and $\ell = \ell_{\text{global}}$.

**Lemma A.1.** *[proof in page 24] Let $(\mathcal{X}, f_X, M, \Theta, \Phi, \ell)$ be a global discrimination game. Assuming $\Phi$ is unrestricted, a sender $S_\theta$ is optimal if and only if it minimizes the following objective:*

$$-I(X; S_\theta(X))$$

We see that optimal protocols maximize mutual information between inputs and messages. This corresponds to a popular idea in representation learning literature [19, 38]. However, similarly to the objective shown in Lemma 5.3 for the regular discrimination game, this communication goal does not consider distances between inputs. In fact, mutual information can be calculated for non-numerical input spaces.

## A.2 Supervised discrimination

In this variation, labels are incorporated into the game via the candidate choice mechanism. The distractors (candidates that are not the target) are guaranteed to not have the same label as the target. This means that the receiver will never have to tell apart two inputs from the same class.

Let $\mathcal{Y}$ be a finite set of labels, and let $\text{label} \colon \mathcal{X} \to \mathcal{Y}$ be a (deterministic) function mapping input to label. Denote the distribution over labels $Y = \text{label}(X)$. We assume that this random variable has a uniform distribution, i.e., that the classes are balanced. The supervised variation uses the same receiver functions as in Definition 2, and a slightly different loss function:

$$\ell_{\text{supervised}}^{X,Y,d}(\theta, \varphi, x) := \mathbb{E}_{\substack{t \sim U\{1,\ldots,d\},\ x_t = x \\ x_1,\ldots,x_{(t-1)}, x_{(t+1)},\ldots,x_d \sim P(X|Y \neq \text{label}(x))^{d-1}}} -\log P(R_\varphi(S_\theta(x), x_1, \ldots, x_d) = t)$$

**Definition 8.** For $d \leq |\mathcal{Y}|$, an EC setup $(\mathcal{X}, f_X, M, \Theta, \Phi, \ell)$ is a *d-candidates supervised discrimination game* if $\Phi \subseteq \Phi_{\text{discrimination}}^{M,\mathcal{X},d}$ and $\ell = \ell_{\text{supervised}}^{X,Y,d}$.

**Lemma A.2.** *[proof in page 24] Let $(\mathcal{X}, f_X, M, \Theta, \Phi, \ell)$ be a 2-candidates supervised discrimination game. Assuming $\Phi$ is unrestricted, a sender $S_\theta$ is optimal if and only if it minimizes the following objective:*

$$\sum_{m \in M} P(S_\theta(X) = m)^2 - \sum_{m \in M} \sum_{y \in \mathcal{Y}} P(S_\theta(X) = m, label(X) = y)^2$$

This result is interpretable too. The first expression is the unsupervised objective that encourages uniformity of message probabilities, which we encountered in Lemma 5.3. The second expression encourages non-diversity of labels within each equivalence class (optimal if all inputs mapped to a message share the same label). This means that optimal messages would have high mutual information with the labels, which in turn could introduce semantic consistency. However, within each class, the communication protocol would not necessarily be able to distinguish between inputs; perfect accuracy can be achieved with just $|\mathcal{Y}|$ unique messages.

This result presents another key implication, regarding complexity in the discrimination game [15]. We can consider a candidate choice mechanism that makes sure all distractors have the same label as the target (the opposite of the classic supervised game). This task will require the receiver to tell apart similar inputs, resulting in high contextual complexity. Interestingly, this task incentivizes the sender to map same-label inputs to different messages, in sharp contrast from Lemma A.2. Therefore, messages learned to solve this high-complexity task will have **low mutual information** with the labels, i.e., this game is likely to result in counter-intuitive communication protocols.

### A.3 Classification discrimination

The classification variation builds on top of the supervised version, by swapping out the target from the set of candidates, and replacing it with a random input that has the same label. In other words, the receiver attempts to guess the candidate with the same label as the target. This game is structured such that every class has exactly one representation in the set of candidates. This is somewhat related to the object-focused referential game [10, 12].

Let $\mathcal{Y} = \{1, \ldots, n\}$ be a finite set of labels, and let label : $\mathcal{X} \to \mathcal{Y}$ be a (deterministic) function mapping input to label. Denote the distribution over labels $Y = \text{label}(X)$. The classification variation uses the same receiver functions as in Definition 2, and the following loss function:

$$\ell_{\text{classification}}^{X,Y}(\theta, \varphi, x) := \mathop{\mathbb{E}}_{\{x_i \sim P(X|Y=i)\}_{i=1}^n} -\log P(R_\varphi(S_\theta(x), x_1, \ldots, x_n) = \text{label}(x))$$

**Definition 9.** An EC setup $(\mathcal{X}, f_X, M, \Theta, \Phi, \ell)$ is a *classification discrimination game* if $\Phi \subseteq \Phi_{\text{discrimination}}^{M,\mathcal{X},n}$ and $\ell = \ell_{\text{classification}}^{X,Y}$.

Note that if $\mathcal{X}$ is finite, we could choose $n = |\mathcal{X}|$ and $\text{label}(x_i) = i$, i.e. define labels as the identity mapping, and end up with the global discrimination game.

**Lemma A.3.** *[proof in page 25] Let $(\mathcal{X}, f_X, M, \Theta, \Phi, \ell)$ be a classification discrimination game. Assuming $\Phi$ is unrestricted, a sender $S_\theta$ is optimal if and only if it minimizes the following objective:*

$$-I(Y; S_\theta(X))$$

Optimal communication protocols in the classification game maximize mutual information with the labels. If the labels signal meaningful properties of the inputs, protocol induced by this game will reflect that, and be semantically consistent. However, similarly to the supervised discrimination game, optimal solutions can use just $|\mathcal{Y}|$ different messages, as they do not have to describe any properties of the input other than its label.

## B Reconstruction and k-means clustering

Lemma 5.1 shows that optimal communication protocols in the reconstruction setting minimize the objective function of k-means clustering [36], where each equivalence class corresponds to a cluster. In this section we show a stronger connection between the two environments. Namely, we show that if $X$ is a uniform distribution with finite support, both $\Theta$ and $\Phi$ are unrestricted, and the message space $M$ is finite, the reconstruction game can simulate k-means clustering with $k = |M|$ (the equivalence classes correspond to clusters). This happens if we alternate between each agent's optimization:

**Assignment step** When the receiver is frozen and sender is optimized, the best solution (synchronized sender) assigns each input to its projection on Receiver's Image:

$$R(S^*(x)) = R(\arg\min_{m \in M} \|x - R(m)\|) = \arg\min_{x' \in \text{Img}(R)} \|x - x'\|.$$

In other words, the sender chooses the message (cluster) with the closest receiver output (centroid) to its input.

**Update step** When the sender is frozen and receiver is optimized, the best solution (synchronized receiver) sets each output to be the mean over the relevant inputs:

$$R^*(m) = \arg\min_{r \in \mathcal{X}} \mathop{\mathbb{E}}_{x \sim I} \left[ \|r - x\|^2 \mid S(x) = m \right] = \mathbb{E}\left[X \mid S(X) = m\right]$$

In other words, the receiver updates its output (centroid) to be the mean over inputs mapped to the given message (cluster).

## C Proofs

For ease of notation, we often write $S, S^*, R, R^*$ instead of $S_\theta, S_{\theta^*}, R_\varphi, R_{\varphi^*}$ when the hypotheses are clear from context. Additionally, we use square brackets as a simple notation of a message's equivalence class, i.e.,

$$[m] \equiv [m]_\theta := S_\theta^{-1}(m) = \{x \in \mathcal{X} : S_\theta(x) = m\}$$

### C.1 Simplification of the semantic consistency definition

*simplification of Equation* (1). Recall Equation (1):

$$\underset{x_1,x_2 \sim X}{\mathbb{E}}\left[\|x_1 - x_2\|^2 \mid S_\theta(x_1) = S_\theta(x_2)\right] < \underset{x_1,x_2 \sim X}{\mathbb{E}}\left[\|x_1 - x_2\|^2\right]$$

Note that since $\mathcal{X}$ is bounded, these values as well as the expectation and variance of $X$ are finite. Using the identity $\underset{x_1,x_2 \sim X}{\mathbb{E}}[\|x_1 - x_2\|^2] = 2\mathrm{Var}[X]$, the left term is equal to

$$\underset{x_1,x_2 \sim X}{\mathbb{E}}\left[\|x_1 - x_2\|^2 \mid S_\theta(x_1) = S_\theta(x_2)\right]$$

$$= \underset{m \sim S_\theta(X)}{\mathbb{E}} \underset{x_1,x_2 \sim X}{\mathbb{E}}\left[\|x_1 - x_2\|^2 \mid S_\theta(x_1) = S_\theta(x_2) , \ S_\theta(x_1) = m\right]$$

$$= \underset{m \sim S_\theta(X)}{\mathbb{E}} \underset{x_1,x_2 \sim X}{\mathbb{E}}\left[\|x_1 - x_2\|^2 \mid S_\theta(x_1) = m, \ , S_\theta(x_2) = m\right]$$

$$= 2 \cdot \underset{m \sim S_\theta(\mathcal{X})}{\mathbb{E}}\left[\mathrm{Var}\left[X \mid S_\theta(X) = m\right]\right]$$

and the right term is simply $2\mathrm{Var}[X]$. Applying this to Equation (1) yields Definition 3. $\qquad\square$

### C.2 Semantic consistency proofs

**Reconstruction**

*Lemma 5.1.* Let $(\mathcal{X}, f_X, M, \Theta, \Phi, \ell)$ be a reconstruction game. Assuming $\Phi$ is unrestricted, a sender $S_\theta$ is optimal if and only if it minimizes the following objective:

$$\sum_{m \in M} P(S_\theta(X) = m) \cdot \mathrm{Var}\left[X \mid S_\theta(X) = m\right]$$

*Proof of Lemma 5.1.* Let $X_{[m]}$ denote the distribution $P\left(X \mid X \in [m]\right)$.
The synchronized receiver for sender $S$ is defined by:

$$R^* = \underset{R \in \Phi}{\arg\min} \ \underset{x \sim X}{\mathbb{E}}\|R(S(x)) - x\|^2$$

Since $\Phi$ is assumed to be unrestricted, we get

$$R^*(m) = \underset{r \in \mathcal{X}}{\arg\min} \ \underset{x \sim X}{\mathbb{E}}\left[\|r - x\|^2 \mid S(x) = m\right]$$

$$= \underset{r \in \mathcal{X}}{\arg\min} \ \underset{x \sim X_{[m]}}{\mathbb{E}}\|r - x\|^2$$

$$= \mathbb{E}\left[X_{[m]}\right]$$

We can apply this synchronized receiver back in the reconstruction loss function without changing the set of optimal sender hypotheses, since a sender is optimal if and only if when paired with a

synchronized receiver, the pair is optimal. Note that $[m] = S^{-1}(m)$ depends on sender. We get:

$$\begin{aligned}
S^* &\in \arg\min_{S \in \Theta} \; \mathbb{E}_{x \sim X} \left\| R^*(S(x)) - x \right\|^2 \\
&= \arg\min_{S \in \Theta} \; \mathbb{E}_{x \sim X} \left\| \mathbb{E}\left[ X_{[S(x)]} \right] - x \right\|^2 \\
&= \arg\min_{S \in \Theta} \sum_{m \in M} \left[ P(X \in [m]) \cdot \mathbb{E}_{x \sim X_{[m]}} \left\| \mathbb{E}\left[ X_{[S(i)]} \right] - x \right\|^2 \right] \\
&= \arg\min_{S \in \Theta} \sum_{m \in M} \left[ P(X \in [m]) \cdot \mathbb{E}_{x \sim X_{[m]}} \left\| \mathbb{E}\left[ X_{[m]} \right] - x \right\|^2 \right] \\
&= \arg\min_{S \in \Theta} \sum_{m \in M} P(X \in [m]) \cdot \mathrm{Var}\left[ X_{[m]} \right]
\end{aligned}$$

The third line is the law of total expectation, where the events are the assignment to message $m$. □

*Theorem 5.2.* Let $(\mathcal{X}, f_X, M, \Theta, \Phi, \ell)$ be a reconstruction game. Assuming $\Phi$ is unrestricted and $\Theta$ contains at least one semantically consistent protocol, every optimal communication protocol is semantically consistent.

*Proof of Theorem 5.2.* Following Lemma 5.1, let $\theta^*$ be an optimal reconstruction sender, i.e.,

$$\begin{aligned}
\theta^* &\in \arg\min_{\theta} \sum_{m \in M} P(X \in [m]_\theta) \cdot \mathrm{Var}\left[ X \mid X \in [m]_\theta \right] \\
&= \arg\min_{\theta} \; \mathbb{E}\left[ \mathrm{Var}\left[ X \mid S_\theta(X) \right] \right] \\
&= \arg\max_{\theta} \; -\mathbb{E}\left[ \mathrm{Var}\left[ X \mid S_\theta(X) \right] \right] \\
&= \arg\max_{\theta} \; \mathrm{Var}\left[ X \right] - \mathbb{E}\left[ \mathrm{Var}\left[ X \mid S_\theta(X) \right] \right]
\end{aligned}$$

The final formula is non-negative by variance decomposition. Since $\Theta$ contains a semantically consistent protocol, this expressions can be greater than zero, so since $\theta^*$ maximizes it, we get

$$\mathrm{Var}\left[ X \right] - \mathbb{E}\left[ \mathrm{Var}\left[ X \mid S_\theta(X) \right] \right] > 0$$

Which proves that Definition 3 is satisfied. □

**Discrimination**

*Lemma 5.3.* Let $(\mathcal{X}, f_X, M, \Theta, \Phi, \ell)$ be a $d$-candidates discrimination game. Assuming $\Phi$ is unrestricted, a sender $S_\theta$ is optimal if and only if it minimizes the following objective:

$$\sum_{m \in M} P(S_\theta(X) = m) \cdot \mathbb{E} \, \log \left( 1 + \mathrm{Binomial}\big( d - 1, P(S_\theta(X) = m) \big) \right)$$

And when $d = 2$ (a single-distractor game), this simplifies into

$$\sum_{m \in M} P(S_\theta(X) = m)^2$$

*Proof of Lemma 5.3.* Given message $m$ and candidates $(x_1, \ldots, x_d)$, such that $t$ is the index of the target $(S(x_t) = m)$, denote $C = \{x_1, \ldots, x_d\} \cap [m]$. Since $t$ is sampled uniformly and only candidates in $C$ have a non-zero likelihood of being the target, we have for all $j \in \{1, \ldots, d\}$

$$\begin{aligned}
P\left( t = j \mid x_1, \ldots, x_d, m \right) &= \frac{f_X(x_1, \ldots, x_d) \cdot P(t = j \mid x_1, \ldots, x_d) \cdot P(S(x_t) = m \mid x_1, \ldots, x_d, t = j)}{f_X(x_1, \ldots, x_d) \cdot P(S(x_t) = m \mid x_1, \ldots, x_d)} \\
&= \frac{\frac{1}{d} \cdot \mathbf{1}\{S(x_j) = m\}}{\frac{|C|}{d}} \\
&= \frac{1}{|C|} \cdot \mathbf{1}\{x_j \in C\}
\end{aligned}$$

Denote this probability $p_j$. A synchronized receiver for sender $S$ is defined as

$$R^* \in \underset{R \in \Phi}{\arg\min} \ \underset{x \sim X}{\mathbb{E}} \ \ell(\theta, R, x)$$

$$= \underset{R \in \Phi}{\arg\min} \ \underset{x \sim X}{\mathbb{E}} \ \underset{\substack{t \sim U\{1,\dots,d\}, \ x_t = x \\ x_1,\dots,x_{(t-1)},x_{(t+1)},\dots,x_d \sim X^{d-1}}}{\mathbb{E}} \ - \log P(R(S(x), x_1, \dots, x_d) = t)$$

$$= \underset{R \in \Phi}{\arg\min} \ \underset{\substack{x_1,\dots,x_d \sim X \\ t \sim U\{1,\dots,d\}}}{\mathbb{E}} \ - \log P(R(S(x_t), x_1, \dots, x_d) = t)$$

Since $\Phi$ is assumed to be unrestricted, we get

$$R^*(m, x_1, \dots, x_d) \in \underset{(q_1,\dots,q_d) \in \Delta\{1,\dots,d\}}{\arg\min} \ \underset{t \sim U\{1,\dots,d\}}{\mathbb{E}} \ [-\log q_t \mid x_1, \dots, x_d, m, S(x_t) = m]$$

$$= \underset{(q_1,\dots,q_d) \in \Delta\{1,\dots,d\}}{\arg\min} \ - \sum_{j=1}^{d} p_j \log q_j$$

A lower bound on the objective (Gibbs' inequality):

$$- \sum_{j=1}^{d} p_j \log q_j \geq - \sum_{j=1}^{d} p_j \log p_j$$

This lower bound is achieved by selecting $q_j = p_j$ (negative log-likelihood is a proper scoring rule). Note that $p_j \geq 0 \ \forall j$ and $\sum_{j=1}^{d} p_j = 1$, making it a legal output for $R$. We now apply this receiver back in the loss function, similarly to the proof of Lemma 5.1. Recall that $x_1, \dots, x_d$ are the candidates, and given the target's index $t$, the other candidates $(x_1, \dots, x_{(t-1)}, x_{(t+1)}, \dots, x_d)$ are called distractors. Also note that $[m] = S^{-1}(m)$ depends on sender.

$$S^* \in \underset{S \in \Theta}{\arg\min} \ \underset{x \sim X}{\mathbb{E}} \ \ell(S, R^*, x)$$

$$= \underset{S \in \Theta}{\arg\min} \ \underset{\substack{x_1,\dots,x_d \sim X \\ t \sim U\{1,\dots,d\}}}{\mathbb{E}} \ - \log P(R^*(S(x_t), x_1, \dots, x_d) = t)$$

$$= \underset{S \in \Theta}{\arg\min} \ \underset{\substack{x_1,\dots,x_d \sim X \\ t \sim U\{1,\dots,d\}}}{\mathbb{E}} \ - \log \frac{1}{|\{x_1, \dots, x_d\} \cap [S(x_t)]|}$$

$$= \underset{S \in \Theta}{\arg\min} \ \underset{\substack{x_1,\dots,x_d \sim X \\ t \sim U\{1,\dots,d\}}}{\mathbb{E}} \ \log |\{x_1, \dots, x_d\} \cap [S(x_t)]|$$

$$= \underset{S \in \Theta}{\arg\min} \ \sum_{m \in M} P(X \in [m]) \ \underset{\substack{x_1,\dots,x_d \sim X \\ t \sim U\{1,\dots,d\}}}{\mathbb{E}} \ \left[ \log |\{x_1, \dots, x_d\} \cap [S(x_t)]| \ \Big| \ S(x_t) = m \right]$$

$$= \underset{S \in \Theta}{\arg\min} \ \sum_{m \in M} P(X \in [m]) \ \underset{\substack{x_1,\dots,x_d \sim X \\ t \sim U\{1,\dots,d\}}}{\mathbb{E}} \ \log \left( 1 + |\{x_1, \dots, x_{(t-1)}, x_{(t+1)}, \dots, x_d\} \cap [m]| \right)$$

$$= \underset{S \in \Theta}{\arg\min} \ \sum_{m \in M} P(X \in [m]) \ \underset{\text{distractors} \sim X^{(d-1)}}{\mathbb{E}} \ \log \left( 1 + |\text{distractors} \cap [m]| \right)$$

$$= \underset{S \in \Theta}{\arg\min} \ \sum_{m \in M} P(X \in [m]) \cdot \mathbb{E} \ \log \left( 1 + \text{Binomial}(d - 1, P(X \in [m])) \right)$$

if $d = 2$ (a single-distractor game), the expression simplifies to:

$$\underset{S \in \Theta}{\arg\min} \ \sum_{m \in M} P(X \in [m]) \cdot \mathbb{E} \ \log \left( 1 + \text{Binomial}(1, P(I \in [m])) \right)$$

$$= \underset{S \in \Theta}{\arg\min} \ \sum_{m \in M} P(X \in [m]) \cdot \left( P(X \in [m]) \cdot \log 2 + (1 - P(X \in [m])) \cdot \log 1 \right)$$

$$= \underset{S \in \Theta}{\arg\min} \ \sum_{m \in M} P(X \in [m])^2$$

$\square$

*Corollary 5.4.* If the set of possible messages is finite, any sender agent that assigns them such that their distribution is uniform can be paired with some receiver to generate globally optimal loss for the discrimination game.

*Proof of Corollary 5.4.* Let $n = |M|$ be the finite number of available messages. Denote the vector $\mathbf{p} = (p_1, \ldots, p_n)$ such that $p_i = P(X \in [m_i])$. For $x \in [0, 1]$, denote the function

$$f(x) = x \cdot \mathbb{E} \, \log(1 + \text{Binomial}(d - 1, x))$$

Following Lemma 5.3, we can formulate the task as a constrained optimization problem on the simplex:

$$\min_{\mathbf{p} \in \mathbb{R}^n} \quad F(\mathbf{p}) := \sum_{i=1}^{n} f(p_i)$$
$$\text{s.t.} \quad \mathbf{p}^T \mathbf{1} = 1$$
$$\mathbf{p} \geq 0$$

Note that the distribution of $X$ may create additional constraints. We will prove that $f$ is convex, meaning that this is a convex problem, so any KKT solution is global minimum [26]. The uniform solution $\mathbf{p}^* = (\frac{1}{n}, \ldots, \frac{1}{n})$ is indeed KKT: assigning $\lambda = \vec{0}$ and $\mu = -\nabla f(\frac{1}{n})$ we have

$$\nabla F(\mathbf{p}^*) + \lambda^T \nabla(-\mathbf{p})(\mathbf{p}^*) + \mu \cdot \nabla(\mathbf{p}^T \mathbf{1} - 1)(\mathbf{p}^*) = \nabla f\left(\frac{1}{n}\right) + \lambda^T(-I_n) + \mu \cdot \mathbf{1} = \vec{0}$$

It is left to prove that $f$ is convex over $[0, 1]$. For simplicity, we replace $d - 1$ with $d \in \mathbb{N}^+$:

$$f(x) = x \cdot \mathbb{E}[\log(1 + \text{Binom}(d, x))]$$

This convexity proof is adjusted from mathoverflow (link). Denote

$$\tilde{g}(k) = \log(1 + k)$$

and

$$B_g^d(x) = \mathbb{E}[g(\text{Binom}(d, x))] = \sum_{k=0}^{d} g(k) \binom{d}{k} x^k (1 - x)^{d-k} \quad \Rightarrow \quad f(x) = x \cdot B_g^d(x)$$

We have

$$B_g^{d\prime}(x) = \sum_{k=0}^{d} g(k) \binom{d}{k} k x^{k-1} (1 - x)^{d-k} - \sum_{k=0}^{d} g(k) \binom{d}{k} (d - k) x^k (1 - x)^{d-k-1}$$

$$= \sum_{k=1}^{d} g(k) \binom{d}{k} k x^{k-1} (1 - x)^{d-k} - \sum_{k=0}^{d-1} g(k) \binom{d}{k} (d - k) x^k (1 - x)^{d-k-1}$$

Using the identities $\binom{d}{k} k = d\binom{d-1}{k-1}$ and $\binom{d}{k}(d - k) = d\binom{d-1}{k}$, we get

$$B_g^{d\prime}(x) = \sum_{k=1}^{d} g(k) d \binom{d-1}{k-1} x^{k-1} (1 - x)^{d-k} - \sum_{k=0}^{d-1} g(k) d \binom{d-1}{k} x^k (1 - x)^{d-k-1}$$

$$= d \sum_{k=0}^{d-1} g(k+1) \binom{d-1}{k} x^k (1 - x)^{d-1-k} - d \sum_{k=0}^{d-1} g(k) \binom{d-1}{k} x^k (1 - x)^{d-1-k}$$

$$= d \sum_{k=0}^{d-1} (g(k+1) - g(k)) \binom{d-1}{k} x^k (1 - x)^{d-1-k}$$

$$= d \cdot B_{\Delta g}^{d-1}(x)$$

where $\Delta g(k) = g(k+1) - g(k)$.

We can apply this again to get the second derivative of $B_g^d$:

$$B_g^{d\prime\prime}(x) = d \cdot B_{\Delta g}^{d-1\prime}(x) = d(d-1) B_{\Delta^2 g}^{d-2}(x)$$

We can now write the second derivative of $f$:

$$f''(x) = 2B_{\tilde{g}}^{d'}(x) + x \cdot B_{\tilde{g}}^{d''}(x)$$

$$= 2d \cdot B_{\Delta\tilde{g}}^{d-1}(x) + x \cdot d(d-1)B_{\Delta^2\tilde{g}}^{d-2}(x)$$

$$= 2d\sum_{k=0}^{d-1}\Delta\tilde{g}(k)\binom{d-1}{k}x^k(1-x)^{d-1-k} + d(d-1)\sum_{k=0}^{d-2}\Delta^2\tilde{g}(k)\binom{d-2}{k}x^{k+1}(1-x)^{d-2-k}$$

Since $\binom{d-2}{k-1} = \frac{k}{d-1}\binom{d-1}{k}$, the second term can be written as

$$d(d-1)\sum_{k=0}^{d-2}\Delta^2\tilde{g}(k)\binom{d-2}{k}x^{k+1}(1-x)^{d-2-k} = d(d-1)\sum_{k=1}^{d-1}\Delta^2\tilde{g}(k-1)\binom{d-2}{k-1}x^k(1-x)^{d-1-k}$$

$$= d\sum_{k=1}^{d-1}\Delta^2\tilde{g}(k-1)k\binom{d-1}{k}x^k(1-x)^{d-1-k}$$

$$= d\sum_{k=0}^{d-1}\Delta^2\tilde{g}(k-1)k\binom{d-1}{k}x^k(1-x)^{d-1-k}$$

Thus, we have

$$f''(x) = 2d\sum_{k=0}^{d-1}\Delta\tilde{g}(k)\binom{d-1}{k}x^k(1-x)^{d-1-k} + d\sum_{k=0}^{d-1}\Delta^2\tilde{g}(k-1)k\binom{d-1}{k}x^k(1-x)^{d-1-k}$$

$$= d\sum_{k=0}^{d-1}\left(2\Delta\tilde{g}(k) + \Delta^2\tilde{g}(k-1)k\right)\binom{d-1}{k}x^k(1-x)^{d-1-k}$$

To finish the proof, we show that the term $h(k) \equiv 2\Delta\tilde{g}(k) + \Delta^2\tilde{g}(k-1)k$ is positive, proving that $f$ is convex.

$$\Delta^2 g(k) = \Delta g(k+1) - \Delta g(k) = g(k+2) - g(k+1) - (g(k+1) - g(k))$$
$$= g(k+2) + g(k) - 2g(k+1)$$
$$2\Delta g(k) + \Delta^2 g(k-1)k = 2g(k+1) - 2g(k) + k(g(k+1) + g(k-1) - 2g(k))$$
$$= (k+2)\cdot g(k+1) + k\cdot g(k-1) - 2(k+1)\cdot g(k)$$

We get

$$h(k) \equiv 2\Delta\tilde{g}(k) + \Delta^2\tilde{g}(k-1)k = (k+2)\cdot\log(k+2) + k\cdot\log(k) - 2(k+1)\cdot\log(k+1)$$

For $k > 0$, the function $t\log t$ is convex, thus $h(k) > 0$ by Jensen's inequality. Also, $h(0) = 2\log 2 > 0$.

$\square$

*Theorem 5.5.* There exists a discrimination game $(\mathcal{X}, f_X, M, \Theta, \Phi, \ell)$ where $\Phi$ is unrestricted and $\Theta$ contains at least one semantically consistent protocol, in which not all of the optimal communication protocols are semantically consistent.

*Proof of Theorem 5.5.* If $X$ is a uniform distribution over a finite support, then as shown by Corollary 5.4, a Sender is optimal if it maps the same number of inputs to every message. Thus, for example, if $|\mathcal{X}| = 2 \cdot |M|$ (both finite), any Sender that maps exactly two inputs to each message is optimal, and we can define such an agent by mapping the least similar pairs of inputs together, creating an optimal Sender that does not induce a meaning-consistent latent space. For a concrete example, see the proof for Theorem 6.2. $\square$

## C.3 Spatial meaningfulness proofs

*Theorem 6.1.* Let $(\mathcal{X}, f_X, M, \Theta, \Phi, \ell)$ be a reconstruction game, let $\varepsilon_0 \geq \varepsilon_M$ and let $\varphi \in \Phi$ such that $R_\varphi$ is $(X, M, \varepsilon_0)$-simple and non-degenerate. Every sender that is synchronized with $R_\varphi$ is $\varepsilon_0$-spatially meaningful.

*Proof of Theorem 6.1.* In the reconstruction task, the optimal constant receiver is

$$
\begin{aligned}
C^* &= \arg\min_{x \sim X} \mathbb{E}_{x \sim X} \ell(\theta, \varphi^C, x) \\
&= \arg\min \mathbb{E}_{x \sim X} \|C - x\|^2 \\
&= \mathbb{E}_{x \sim X}[x] \equiv \bar{X}
\end{aligned}
$$

Let $\theta^*$ be a sender synchronized with $R_\varphi$. Denote $L = \sup_{x \in \mathcal{X}} \ell(\theta^*, \varphi, x)$ and $\bar{L} = \frac{1}{2} \cdot \mathbb{E}_{x \sim X} \|x - \bar{X}\|^2$. Since $R_\varphi$ is strictly better than constant, we have

$$
\forall x \in \mathcal{X} : \quad \ell(\theta^*, \varphi, x) \leq L \leq \frac{\bar{L}}{2}
$$

Denote the receiver's prediction $R_\varphi(S_{\theta^*}(x)) := \hat{x}_\varphi$. Note that when the sender's hypothesis class $\Theta$ is unrestricted, this becomes the projection on $R_\varphi$'s image:

$$
R_\varphi(S_{\theta^*}(x)) = R_\varphi(\arg\min_{m \in M} \|x - R_\varphi(m)\|) = \arg\min_{x' \in \mathrm{Img}(R_\varphi)} \|x - x'\| = \mathrm{Proj}_{\mathrm{Img}(R_\varphi)}(x).
$$

By the definition of $\ell_{\text{reconstruction}}$, we get

$$
\begin{aligned}
\forall x \in \mathcal{X} \quad & \|x - \hat{x}_\varphi\|^2 \leq L \\
\Rightarrow \quad \forall x \in \mathcal{X} \quad & \|x - \hat{x}_\varphi\| \leq \sqrt{L}
\end{aligned}
$$

Let $l_R = \frac{\sqrt{2}-1}{2\varepsilon_0} \sqrt{\mathrm{Var}[X]}$ be the constant from Definition 5. For any $\varepsilon > 0$ and $x_1, x_2 \in \mathcal{X}$ such that

$$
\|S_{\theta^*}(x_1) - S_{\theta^*}(x_2)\| \leq \varepsilon
$$

we have, since $R_\varphi$ is $(X, M, \varepsilon_0)$-simple, that

$$
\|\hat{x}_{1\varphi} - \hat{x}_{2\varphi}\| = \|R_\varphi(S_{\theta^*}(x_1)) - R_\varphi(S_{\theta^*}(x_2))\| \leq l_R \cdot \varepsilon
$$

Thus:

$$
\|x_1 - x_2\| \leq \|x_1 - \hat{x}_{1\varphi}\| + \|\hat{x}_{1\varphi} - \hat{x}_{2\varphi}\| + \|x_2 - \hat{x}_{2\varphi}\| \leq 2\sqrt{L} + l_R \cdot \varepsilon
$$

We now show that every $\varepsilon \leq \varepsilon_0$ holds $2\sqrt{L} + l_R \cdot \varepsilon < 2\sqrt{\bar{L}}$. First, note that

$$
\varepsilon_0 = \frac{\frac{\sqrt{2}-1}{2} \sqrt{Var[X]}}{l_R} = \frac{(1 - \frac{1}{\sqrt{2}})\sqrt{\frac{1}{2}Var[X]}}{l_R}
$$

In addition, since $L \leq \frac{\bar{L}}{2}$,

$$
\sqrt{\bar{L}} - \sqrt{L} \geq \sqrt{\bar{L}} - \sqrt{\frac{\bar{L}}{2}} = (1 - \frac{1}{\sqrt{2}})\sqrt{\bar{L}}
$$

so for any $\varepsilon \leq \varepsilon_0$, we get:

$$
\varepsilon \leq \varepsilon_0 = \frac{(1 - \frac{1}{\sqrt{2}})\sqrt{\bar{L}}}{l_R} < \frac{2(1 - \frac{1}{\sqrt{2}})\sqrt{\bar{L}}}{l_R} \leq \frac{2(\sqrt{\bar{L}} - \sqrt{L})}{l_R}
$$

Which finally leads to

$$
2\sqrt{L} + l_R \cdot \varepsilon < 2\sqrt{\bar{L}}
$$

Thus, for any $x_1, x_2 \in \mathcal{X}$ that hold $\|S_{\theta^*}(x_1) - S_{\theta^*}(x_2)\| \leq \varepsilon \leq \varepsilon_0$, we get:

$$
\begin{aligned}
\|x_1 - x_2\|^2 &\leq (2\sqrt{L} + l_R \cdot \varepsilon)^2 \\
&< (2\sqrt{\bar{L}})^2 = 4\bar{L} \\
&= 2 \cdot \mathop{\mathbb{E}}_{x \sim X} \|x - \bar{X}\|^2 \\
&= \mathop{\mathbb{E}}_{x \sim X} \|x - \bar{X}\|^2 + \mathop{\mathbb{E}}_{x \sim X} \|x - \bar{X}\|^2 - 2 \cdot \mathop{\mathbb{E}}_{x \sim X} [x - \bar{X}]^T \mathop{\mathbb{E}}_{x \sim X} [x - \bar{X}] \\
&= \mathop{\mathbb{E}}_{x_1, x_2 \sim X} \|(x_1 - \bar{X}) - (x_2 - \bar{X})\|^2 \\
&= \mathop{\mathbb{E}}_{x_1, x_2 \sim X} \|x_1 - x_2\|^2
\end{aligned}
$$

Taking expectation yields the desired result for Definition 4:

$$
\mathop{\mathbb{E}}_{x_1, x_2 \sim X} \left[ \|x_1 - x_2\|^2 \;\middle|\; \|S_\theta(x_1) - S_\theta(x_2)\| \leq \varepsilon \right] < \mathop{\mathbb{E}}_{x_1, x_2 \sim X} \left[ \|x_1 - x_2\|^2 \right]
$$

$\square$

*Theorem 6.2.* There exists a discrimination game $(\mathcal{X}, f_X, M, \Theta, \Phi, \ell)$, $\varepsilon_0 \geq \varepsilon_M$ and a receiver $\varphi \in \Phi$ which is $(X, M, \varepsilon_0)$-simple and non-degenerate, where a synchronized sender matching $R_\varphi$ is not $\varepsilon$-spatially meaningful for any $\varepsilon$.

*Proof of Theorem 6.2.* We propose the following counter example. Let $M = \{1, 2, 3, 4, 5, 6\}$ and let $X$ be the uniform distribution over the finite support set $\{1, \ldots, 6\} \cup \{-1, \ldots, -6\}$. Let $\varepsilon_0 = \varepsilon_M = 1$. We let both hypothesis classes be unrestricted, set the number of candidates $d$ to 2 (a single distractor), and denote the following sets:

$$
\begin{aligned}
A_1 &= \{1, -1\} \\
A_2 &= \{2, -2\} \\
A_3 &= \{3, -3\} \\
A_4 &= \{4, -4\} \\
A_5 &= \{5, -5\} \\
A_6 &= \{6, -6\}
\end{aligned}
$$

We construct a receiver function that optimally tells apart members from different sets, but is unable to differentiate within each one. Formally:

- $R_\varphi$ cannot tell apart inputs within each set:

$$
\forall k \in \{1, \ldots, 6\}, \; \forall x_1, x_2 \in A_k, \; \forall m \in M : \quad R(m, x_1, x_2) = (0.5, 0.5)
$$

- $R_\varphi$ is optimal for inputs from different sets; Given the message $m = k$ where $k \in \{1, \ldots, 6\}$, $R_\varphi$ outputs the true probabilities of each input being the target conditioned on the event $X \in A_k$:

$$
R_\varphi(k, x_1, x_2) = \begin{cases}
(0.5, 0.5) & x_1 \in A_k \text{ and } x_2 \in A_k \\
(1, 0) & x_1 \in A_k \text{ and } x_2 \notin A_k \\
(0, 1) & x_1 \notin A_k \text{ and } x_2 \in A_k
\end{cases}
$$

The largest possible distance between $R_\varphi$'s outputs is $\frac{1}{\sqrt{2}}$, and the minimal distance between messages is $\varepsilon_M = 1$. Thus,

$$
\|R_\varphi(x_1) - R_\varphi(x_2)\| \leq \frac{1}{\sqrt{2}} \cdot 1 \leq \frac{1}{\sqrt{2}} \cdot \|x_1 - x_2\|.
$$

Note that

$$
\frac{\sqrt{2} - 1}{2\varepsilon_0} \cdot \sqrt{Var[X]} = \frac{\sqrt{2} - 1}{2} \cdot \sqrt{\frac{91}{6}} > \frac{1}{\sqrt{2}}
$$

Which means $R_\varphi$ is $(X, M, \varepsilon_0)$-simple. We now show that the sender that maps each input to the index of its set, i.e.,

$$S_{\tilde\theta}(i) = \begin{cases} 1 & \text{if } x \in A_1 \\ 2 & \text{if } x \in A_2 \\ 3 & \text{if } x \in A_3 \\ 4 & \text{if } x \in A_4 \\ 5 & \text{if } x \in A_5 \\ 6 & \text{if } x \in A_6 \end{cases}$$

is synchronized for $R_\varphi$. Let $x \in \mathcal{X}$ belong to set $A_k$. The loss function is bounded by:

$$\ell(\theta, \varphi, x) = \mathop{\mathbb{E}}_{\substack{t \sim U\{1,2\} \\ x_2 \sim X}} \left[ -\log \begin{cases} R_\varphi(S_\theta(i), x, x_2)_1 & \text{if } t = 1 \\ R_\varphi(S_\theta(i), x_2, x)_2 & \text{if } t = 2 \end{cases} \right]$$

$$= P(X \in A_k) \cdot \mathop{\mathbb{E}}_{\substack{t \sim U\{1,2\} \\ x_2 \sim X|_{A_k}}} \left[ -\log \begin{cases} R_\varphi(S_\theta(x), x, x_2)_1 & \text{if } t = 1 \\ R_\varphi(S_\theta(x), x_2, x)_2 & \text{if } t = 2 \end{cases} \right]$$

$$+ P(X \notin A_k) \cdot \mathop{\mathbb{E}}_{\substack{t \sim U\{1,2\} \\ x_2 \sim X|_{X \setminus A_k}}} \left[ -\log \begin{cases} R_\varphi(S_\theta(x), x, x_2)_1 & \text{if } t = 1 \\ R_\varphi(S_\theta(x), x_2, x)_2 & \text{if } t = 2 \end{cases} \right]$$

$$= P(X \in A_k) \cdot \left( -\log \frac{1}{2} \right) + P(X \notin A_k) \cdot \mathop{\mathbb{E}}_{\substack{t \sim U\{1,2\} \\ x_2 \sim X|_{X \setminus A_k}}} \left[ -\log \begin{cases} R_\varphi(S_\theta(i), x, x_2)_1 & \text{if } t = 1 \\ R_\varphi(S_\theta(i), x_2, x)_2 & \text{if } t = 2 \end{cases} \right]$$

$$\geq P(X \in A_k) \cdot (\log 2) + P(X \notin A_k) \cdot (-\log 1)$$

$$= \ell(\tilde\theta, \varphi, x).$$

This means that $\tilde\theta$ is synchronized for $R_\varphi$. In fact, $S_{\tilde\theta}$ is optimal by Corollary 5.4. We can continue simplifying the expression to show that $R_\varphi$ is strictly better than constant:

$$\ell(\tilde\theta, \varphi, x) = P(X \in A_k) \cdot (\log 2) + P(X \notin A_k) \cdot (-\log 1)$$

$$= \frac{1}{6} \cdot \log 2$$

$$< \frac{1}{4} \cdot \log 2$$

And indeed, the average loss of the optimal constant is:

$$\ell(\theta, \varphi^C, i) = \mathop{\mathbb{E}}_{\substack{t \sim U\{1,2\} \\ x_2 \sim I}} \left[ -\log \begin{cases} R_{\varphi^C}(S_\theta(i), x, x_2)_1 & \text{if } t = 1 \\ R_{\varphi^C}(S_\theta(i), x_2, x)_2 & \text{if } t = 2 \end{cases} \right]$$

$$= \mathop{\mathbb{E}}_{\substack{t \sim U\{1,2\} \\ x_2 \sim X}} \left[ -\log \begin{cases} C & \text{if } t = 1 \\ 1 - C & \text{if } t = 2 \end{cases} \right]$$

$$= \frac{1}{2} \cdot (-\log C) + \frac{1}{2} \cdot (-\log(1 - C))$$

$$= -\frac{1}{2} \cdot (\log c + \log(1 - c))$$

$$\nabla \ell(\theta, \varphi^C, x) = -\frac{1}{2} \cdot \left( \frac{1}{C} - \frac{1}{1 - c} \right) \Rightarrow C^* = \frac{1}{2}$$

$$\mathop{\mathbb{E}}_{x \sim X} \ell(\theta, \varphi^{C^*}, x) = -\frac{1}{2} \cdot 2 \cdot \log \frac{1}{2} = \log 2$$

Finally, we can show that $S_{\tilde\theta}$ does not induce a spatially meaningful latent space. We make the observation that

$$\mathbb{E}\left[ X \mid S_{\tilde\theta}(X) = m \right] = 0 \quad \forall\, m \in Img(S_{\tilde\theta})$$

This is a constant function in $m$ which means $\text{Var}(\mathbb{E}\left[X \mid S_{\tilde{\theta}}(X) = m\right]) = 0$. By variance decomposition:

$$\text{Var}\left[X\right] = \mathbb{E}\left(\text{Var}\left[X \mid S_{\tilde{\theta}}(X) = m\right]\right) + \text{Var}\left(\mathbb{E}\left[X \mid S_{\tilde{\theta}}(X) = m\right]\right)$$

$$\text{Var}\left[X\right] = \mathbb{E}\left(\text{Var}\left[X \mid S_{\tilde{\theta}}(X) = m\right]\right)$$

$$\mathbb{E}_{x_1,x_2 \sim X}\left[\|x_1 - x_2\|^2\right] = \mathbb{E}_{m}\, \mathbb{E}_{x_1,x_2 \sim X_{[m]_{\tilde{\theta}}}}\left[\|x_1 - x_2\|^2\right]$$

$$\mathbb{E}_{x_1,x_2 \sim X}\left[\|x_1 - x_2\|^2\right] = \mathbb{E}_{x_1,x_2 \sim X}\left[\|x_1 - x_2\|^2 \mid S_\theta(x_1) = S_\theta(x_2)\right]$$

This equality contradicts Definition 4 that requires strict inequality (for any $\varepsilon_0 > 0$, the requirement will not hold with $\varepsilon = min(\varepsilon_0, 0.9)$.), showing that $S_{\tilde{\theta}}$ is neither meaning consistent nor spatially meaningful. $\square$

### C.4    Proofs for variations of discrimination setups

*Lemma A.1.* Let $(\mathcal{X}, f_X, M, \Theta, \Phi, \ell)$ be a global discrimination game. Assuming $\Phi$ is unrestricted, a sender $S_\theta$ is optimal if and only if it minimizes the following objective:

$$-I(X; S_\theta(X))$$

*Proof of Lemma A.1.* Since negative log-likelihood is a proper scoring rule and $\Phi$ is unrestricted, the synchronized receiver for sender $S$ is

$$R^*(m) = f_X(\,\cdot \mid S(X) = m)$$

Which means that $S^*$ is optimal if and only if:

$$S^* \in \arg\min_{S \in \Theta}\, \mathbb{E}_{x \sim X}\, \ell(S, R^*, x)$$
$$= \arg\min_{S \in \Theta}\, \mathbb{E}_{x \sim X}\, -\log f_X(x \mid S(X) = S(x))$$
$$= \arg\min_{S \in \Theta}\, H(X \mid S(X))$$
$$= \arg\min_{S \in \Theta}\, -I(X; S(X))$$

$\square$

*Lemma A.2.* Let $(\mathcal{X}, f_X, M, \Theta, \Phi, \ell)$ be a 2-candidates supervised discrimination game. Assuming $\Phi$ is unrestricted, a sender $S_\theta$ is optimal if and only if it minimizes the following objective:

$$\sum_{m \in M} P(S_\theta(X) = m)^2 - \sum_{m \in M}\sum_{y \in \mathcal{Y}} P(S_\theta(X) = m, \text{label}(X) = y)^2$$

*Proof of Lemma A.2.* Since negative log-likelihood is a proper scoring rule (shown in the proof of Lemma 5.3) and $\Phi$ is unrestricted, the synchronized receiver for sender $S$ is

$$R^*(m, x_1, x_2) = \mathbb{P}(t \mid x_1, x_2, m)$$

By the game construction, we have

$$P(t = j, x_1, x_2, m) = f_X(x_j) \cdot f_X(x_{3-j} \mid Y \neq \text{label}(x_j)) \cdot \frac{1}{2} \cdot \mathbf{1}\{S(x_j) = m\}$$
$$= \frac{f_X(x_1)f_X(x_2)}{P(Y \neq \text{label}(x_j))} \cdot \frac{1}{2} \cdot \mathbf{1}\{S(x_j) = m\}$$

Since $Y$ is assumed to have uniform distribution, $P(Y \neq \text{label}(x_j)) = \frac{|\mathcal{Y}|-1}{|\mathcal{Y}|}$, which means

$$P(t = j, x_1, x_2, m) = f_X(x_1)f_X(x_2) \cdot \frac{|\mathcal{Y}|}{|\mathcal{Y}| - 1} \cdot \frac{1}{2} \cdot \mathbf{1}\{S(x_j) = m\}$$

Thus, the synchronized receiver is

$$P(R^*(m, x_1, x_2) = j) = P(t = j \mid x_1, x_2, m)$$
$$= \frac{P(t = j, x_1, x_2, m)}{P(t = j, x_1, x_2, m) + P(t = 3 - j, x_1, x_2, m)}$$
$$= \frac{\mathbf{1}\{S(x_j) = m\}}{\mathbf{1}\{S(x_j) = m\} + \mathbf{1}\{S(x_{3-j}) = m\}}$$

Applying this synchronized receiver to the loss function, we get that $S^*$ is optimal if and only if:

$$S^* \in \arg\min_{S \in \Theta} \mathbb{E}_{x \sim X} \ell(S, R^*, x)$$

$$= \arg\min_{S \in \Theta} \mathbb{E}_{\substack{x \sim X \\ x' \sim P(X|Y \neq \text{label}(x))}} \mathbb{E}_{t \sim U\{1,2\}} - \log \left\{ \begin{array}{ll} P(R^*(S(x), x, x') = 1), & \text{if } t = 1 \\ P(R^*(S(x), x', x) = 2), & \text{if } t = 2 \end{array} \right\}$$

$$= \arg\min_{S \in \Theta} \mathbb{E}_{x \sim X} \mathbb{E}_{x' \sim P(X|Y \neq \text{label}(x))} - \log \frac{1}{1 + \mathbf{1}\{S(x') = S(x)\}}$$

$$= \arg\min_{S \in \Theta} \mathbb{E}_{x \sim X} \mathbb{E}_{x' \sim P(X|Y \neq \text{label}(x))} \log(1 + \mathbf{1}\{S(x') = S(x)\})$$

$$= \arg\min_{S \in \Theta} \mathbb{E}_{x \sim X} P(X \notin [S(x)] \mid Y \neq \text{label}(x)) \cdot \log 1 + P(X \in [S(x)] \mid Y \neq \text{label}(x)) \cdot \log 2$$

$$= \arg\min_{S \in \Theta} \mathbb{E}_{x \sim X} P(X \in [S(x)] \mid Y \neq \text{label}(x))$$

$$= \arg\min_{S \in \Theta} \sum_{m \in M} \sum_{y \in \mathcal{Y}} P(X \in [m], Y = y) \cdot \mathbb{E}_{x \sim X} P(X \in [S(x)] \mid Y \neq \text{label}(x), S(X) = m, Y = y)$$

$$= \arg\min_{S \in \Theta} \sum_{m \in M} \sum_{y \in \mathcal{Y}} P(X \in [m], Y = y) \cdot P(X \in [m] \mid Y \neq y)$$

$$= \arg\min_{S \in \Theta} \sum_{m \in M} \sum_{y \in \mathcal{Y}} P(X \in [m], Y = y) \cdot P(X \in [m], Y \neq y)$$

$$= \arg\min_{S \in \Theta} \sum_{m \in M} \sum_{y \in \mathcal{Y}} P(X \in [m], Y = y) \cdot (P(X \in [m]) - P(X \in [m], Y = y))$$

$$= \arg\min_{S \in \Theta} \sum_{m \in M} P(X \in [m]) \sum_{y \in \mathcal{Y}} P(X \in [m], Y = y) - \sum_{m \in M} \sum_{y \in \mathcal{Y}} P(X \in [m], Y = y)^2$$

$$= \arg\min_{S \in \Theta} \sum_{m \in M} P(X \in [m])^2 - \sum_{m \in M} \sum_{y \in \mathcal{Y}} P(X \in [m], Y = y)^2$$

$\square$

*Lemma A.3.* Let $(\mathcal{X}, f_X, M, \Theta, \Phi, \ell)$ be a classification discrimination game. Assuming $\Phi$ is unrestricted, a sender $S_\theta$ is optimal if and only if it minimizes the following objective:

$$-I(Y; S_\theta(X))$$

*Proof of Lemma A.3.* Since negative log-likelihood is a proper scoring rule (shown in the proof of Lemma 5.3) and $\Phi$ is unrestricted, the synchronized receiver for sender $S$ is

$$R^*(m, x_1, x_2) = \mathbb{P}(t \mid x_1, \ldots, x_n, m)$$

By the game construction, we have

$$P(t = j, x_1, \ldots, x_n, m) = P(X \in [m], Y = j) \cdot \prod_{i=1}^{n} f_X(x_i \mid Y = i)$$

Thus, the synchronized receiver is

$$P(R^*(m, x_1, \ldots, x_n) = j) = P(t = j \mid x_1, \ldots, x_n, m)$$
$$= \frac{P(X \in [m], Y = j)}{P(X \in [m])}$$
$$= P(Y = j \mid X \in [m])$$

Applying this synchronized receiver to the loss function, we get that $S^*$ is optimal if and only if:

$$
\begin{aligned}
S^* &\in \underset{S \in \Theta}{\arg\min} \ \underset{x \sim X}{\mathbb{E}} \ \ell(S, R^*, x) \\
&= \underset{S \in \Theta}{\arg\min} \ \underset{x \sim X}{\mathbb{E}} \ \underset{\{x_i \sim P(X|Y=i)\}_{i=1}^n}{\mathbb{E}} \ - \log P(R^*(S(x), x_1, \ldots, x_n) = \text{label}(x)) \\
&= \underset{S \in \Theta}{\arg\min} \ \underset{x \sim X}{\mathbb{E}} \ - \log P(Y = \text{label}(x) \mid X \in [S(x)]) \\
&= \underset{S \in \Theta}{\arg\min} \ H(Y|S(X)) \\
&= \underset{S \in \Theta}{\arg\min} \ -I(Y; S(X))
\end{aligned}
$$

$\square$

# D   Details on empirical work

## D.1   Training details

**Data.**   We use a version of the Shapes dataset [22], where each input is an image of dimensions $(3, 64, 64)$ containing a single object. The object has a random shape out of (square, rectangle, triangle, pentagon, cross, circle, semicircle, ellipse), and a random color out of (red, green, blue, yellow, magenta, cyan, gray). In addition, the object has a random position, rotation, size and distortion. See the Shapes github repo and our code for exact details. We generate $\sim$10K data samples, split into 9088 (142 batches of 64) training samples and 896 (14 batches of 64) validation samples. The MNIST dataset [30] contains $(1, 28, 28)$ images of hand-written digits, split into $\sim 54K$ training samples and 5440 validation samples.

**Agent architectures and loss functions.**   We use the format {Shapes value}/{MNIST value} to indicate values that differ between the datasets. The sender consists of two parts: an *image encoder* and a *message generator*. The image encoder receives an input image of dimensions $(3, 64, 64)/(1, 28, 28)$ and outputs a latent vector of dimension 100. The exact encoder architecture is given in Figure 5a / Figure 6a. A trainable linear layer then maps this output to a vector of dimension 150 which serves as input to the message generator, which is implemented by a single-layer GRU with hidden dimension 150. Its input is used as an initial hidden state, together with a learnable BOS vector. After the GRU cell, the resulting hidden state is mapped by a linear layer to the vocabulary size (10). We use the Gumbel-Softmax relaxation to approximate a symbol sampling, then an embedding matrix is used on the relaxed on-hot vector and the result is fed to the next generation step (along with the current hidden state).

The receiver has different architectures for reconstruction and discrimination. Both receivers start with a *message embedder*, implemented by a single-layer GRU with hidden size 150. In the reconstruction setup, the final hidden state from that network is given to an *image decoder* implemented by transpose-convolutions, see Figure 5b / Figure 6b for full details. The loss in the reconstruction setting is simply the Mean Squared Error (MSE) between the receiver's output and the target image. In the discrimination setting, the receiver uses another image encoder (with the architecture from Figure 5a / Figure 6a) and applies it to every image in the set of candidates. The resulting vector is further processed by a Multilayer perceptron (MLP) with two layers, hidden dimension 150 and relu activation. The output of this MLP is a representation for each candidate. We take a dot product of this representation with the final hidden state of the message embedder (both normalized) to get the score for each candidate. The discrimination loss is then the Cross-Entropy classification loss with the target position as label (also called the infoNCE objective). Visual illustrations of the two tasks are given in Figure 7/Figure 8.

**Training procedure.**   While the architectures described above can be jointly trained from scratch, the discrete optimization with Gumbel-Softmax is much slower and less stable than continuous training. We account for that by employing a two-stage training procedure. In the first stage, we train a continuous autoencoder, which is a concatenation of an image encoder and an image decoder. The trained image encoder is used by the sender agent in both setups, and also by the receiver in the discrimination game. The trained decoder is used by the receiver in the reconstruction game. Both of these pretrained modules are kept frozen during the second stage of training, in which the discrete

game is played out. This means that during EC training, the sender only optimizes the message generator (and a linear layer), and the receiver only optimizes the message embedder (and the MLP in the discrimination setting).

**Training parameters.** We train (during the EC phase) for 200/100 epochs with batch size 64. We use vocabulary size 10 and message length 4. Every message is full length as we do not use a special end-of-sentence token. We use a fixed temperature value of 1.0 for the Gumbel-Softmax sampling method.

**Computation resources.** The main training stage (EC) takes $\sim 4$ hours on Shapes, $\sim 1.5$ hours on MNIST using NVIDIA GeForce RTX 2080 Ti. The continuous training takes $< 10$ minutes.

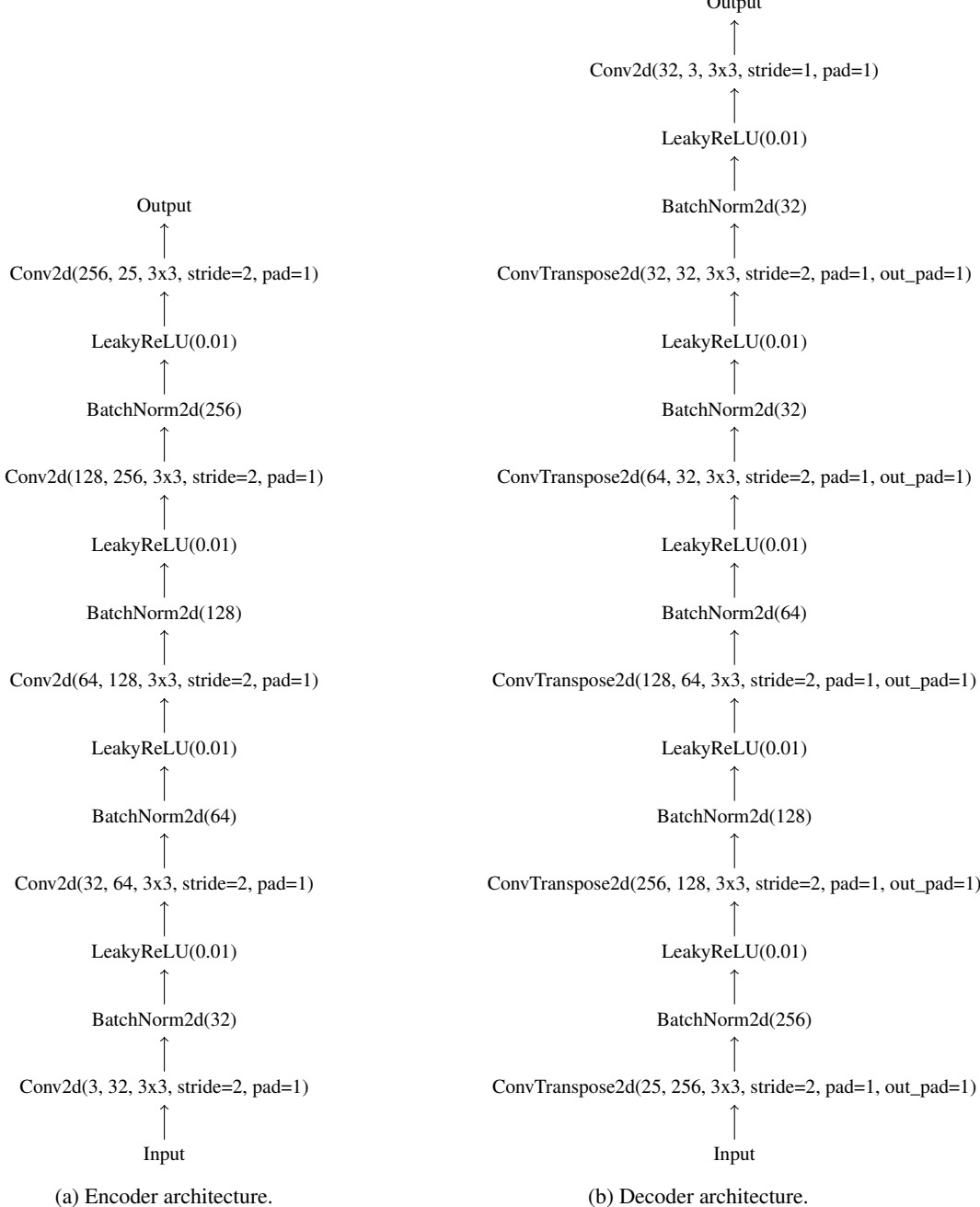

(a) Encoder architecture.

(b) Decoder architecture.

Figure 5: Encoder and Decoder architectures used on Shapes.

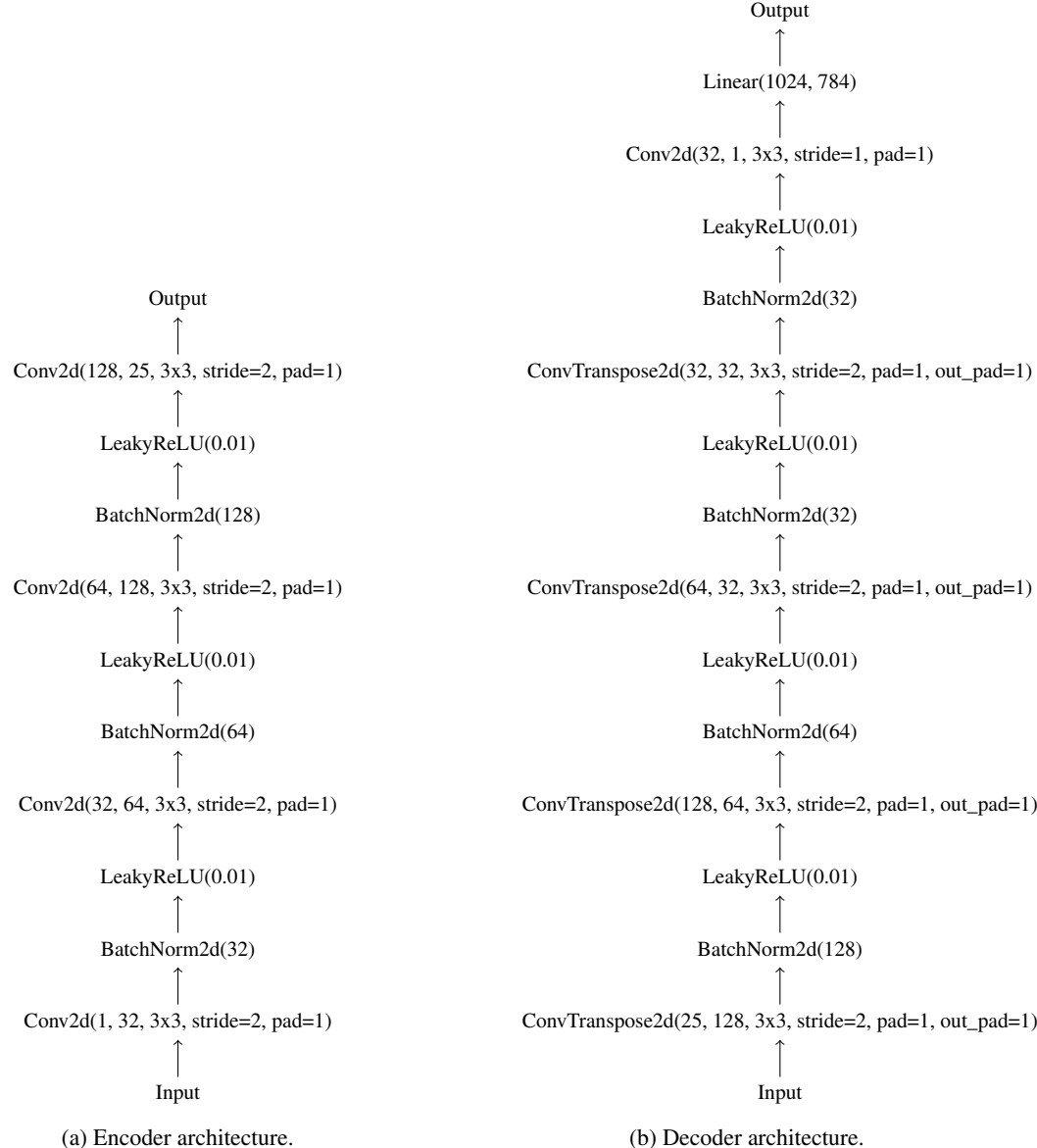

(a) Encoder architecture.

(b) Decoder architecture.

Figure 6: Encoder and Decoder architectures used on MNIST.

## D.2 Metrics

**Discrimination accuracy.** We sample 40 distractors independently, so the receiver predicts the target out of 41 given candidates. Every image in the validation set is used as target exactly once, and the reported accuracy is the precentage of images that were correctly predicted as targets. In the reconstruction setting, the prediction is defined by first passing the given message to the receiver, which outputs a reconstruction of the target, and then picking the candidate which minimizes the loss function (Euclidean distance) with respect to that output.

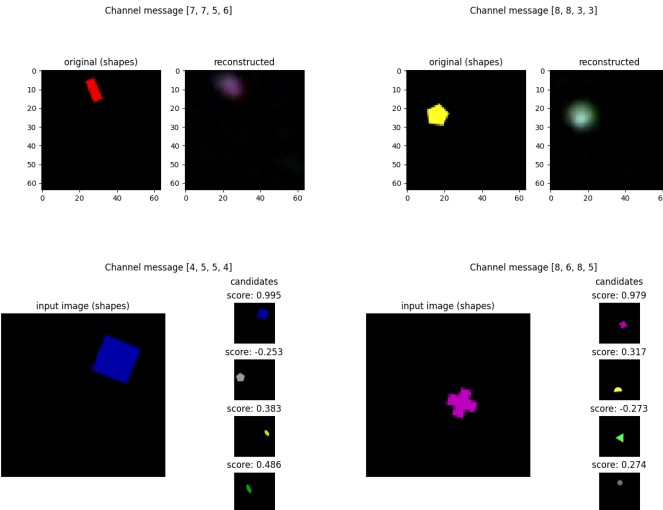

Figure 7: Reconstruction (top) and discrimination (bottom) examples on Shapes

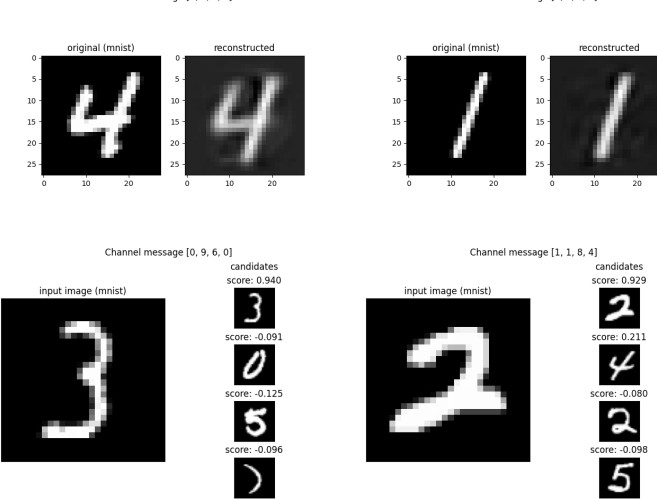

Figure 8: Reconstruction (top) and discrimination (bottom) examples on MNIST

**Message variance.** The unexplained variance is $\mathbb{E}\left[\text{Var}[X \mid S(X)]\right] = \sum_{m \in M} P(m) \cdot \text{Var}\left[[m]\right]$ where $[m]$ is the set of images mapped to message $m$. Using the empirical measures

$$\text{Var}\left[[m]\right] = \frac{1}{2 \cdot |[m]|^2} \sum_{x_1, x_2 \in [m]} \|x_1 - x_2\|^2$$

$$P(m) = \frac{|[m]|}{N}, \qquad N = 896$$

we get

$$\sum_{m \in M} P(m) \cdot \text{Var}\left[[m]\right] = \sum_{m \in M} \frac{|[m]|}{2 \cdot |[m]|^2 \cdot N} \sum_{x_1, x_2 \in [m]} \|x_1 - x_2\|^2$$

$$= \frac{1}{2N} \sum_{m \in M} \frac{1}{|[m]|} \sum_{x_1, x_2 \in [m]} \|x_1 - x_2\|^2$$

We calculate message variance using this formula. A lower value indicates more semantic consistency (less unexplained variance). See Algorithm 1 for the full calculation procedure. Instead of calculating

Euclidean distance between raw pixels [4, 10], which might be uninformative, we use embeddings created by the continuous encoder (perceptual loss [39]).

---

**Algorithm 1** Compute message variance

---

**Require:** Dataset $X$, Set of possible messages $M$, Sender function $S : X \rightarrow M$
1: Initialize dictionary $equiv\_classes$ with keys from $M$ and empty lists as values
2: **for** each input $x \in X$ **do**
3:     $message \leftarrow S(x)$
4:     Append $x$ to $equiv\_classes[message]$
5: **end for**
6: Initialize $pairwise\_sum \leftarrow 0$
7: **for** each $message \in M$ **do**
8:     Initialize $local\_sum \leftarrow 0$
9:     **for** pair $(x_i, x_j)$ in $equiv\_classes[message]$ **do**
10:        $local\_sum \leftarrow local\_sum + \text{distance}(x_i, x_j)$
11:    **end for**
12:    **if** $len(equiv\_classes[message]) > 1$ **then**
13:        $local\_sum \leftarrow local\_sum/len(equiv\_classes[message])$
14:        $pairwise\_sum \leftarrow pairwise\_sum + local\_sum$
15:    **end if**
16: **end for**
17: $N \leftarrow$ size of $X$
18: $result \leftarrow pairwise\_sum/(2 \times N)$
19: **return** $result$

---

**Random baseline.** The distribution of messages induced by the trained sender significantly affects some of the metrics. To facilitate an insightful comparison between the two EC setups, we establish a random baseline per-protocol which randomizes the assignment of images while preserving the number of inputs mapped to each message. Comparing the outcomes across different baselines allows us to examine the extent to which channel utilization impacts the results.

**Topographic similarity.** TopSim [5] is defined as the (Spearman) correlation between pairwise distances in the input space and corresponding distances in the message space. We use edit distance in the message space, and Euclidean distance between embeddings in the input space. The embeddings are generated by the frozen encoder trained in the continuous stage (see Appendix D.1).

**Bag-of-symbols disentanglement.** BosDis [8] calculates per-symbol the gap between the highest and second highest mutual information to an attribute, normalizes by the symbol's entropy and sums over the vocabulary. We consider only two attributes (shape and color), so the mutual information gap is simply the difference between the two.

**Positional disentanglement.** PosDis [8] operates similarly to BosDis, but considers positions within the message rather than symbols.

**Speaker-centered topographic similarity.** S-TopSim [13] operates similarly to PosDis, but rather than summing over positions, the mutual information gap is calculated per attribute. This is a more straightforward application of MIG [9].

**Message purity.** The purity of a message with respect to an attribute is the percentage of images in its equivalence class that have the majority value. The minimal purity value is achieved by a uniform distribution over all of the attribute's possible values; the maximal value, 100%, is reached only if all images mapped to the message share the same value (e.g., if the attribute is shape and all images mapped to that message contain squares). We take a weighted average of message purity over messages. The final measure can be interpreted as the highest attainable accuracy by a deterministic classifier of messages to the attribute. An easier version of this measure, which we term max-purity, defines the purity of each message as the maximum purity over all attributes.

# E  Additional experiments

## E.1  Full results: Compositionality, semantic consistency and game performance

We consider three additional compositionality measures from literature (BosDis, PosDis and S-PosDis), see Appendix D.2 for their definitions. Some of these measures require multiple attributes (based on the mutual information gap), so we report results on Shapes. Table 3 shows these results, and Table 4 shows the correlation between each pair of measures, over three subsets of the collected measurements: the joint set (10 runs), reconstruction only (5 runs) and discrimination only (5 runs). We note that the correlations calculated over the joint set are heavily affected by the two distributions and often show opposite values to the individual tables. Furthermore, the results regarding S-PosDis are somewhat noisy due to its low values reported in Table 3. A key observation is that the correlation between compositionality and game performance (disc. accuracy) is negative or nonexistent across all compositionality measures in both the reconstruction subset and the discrimination subset, other than the S-PosDis value in the discrimination table (which we believe is noise, see previous note). This further corroborates our findings in Section 7.

Table 3: Empirical results with compositionality measures from literature.

|                | TopSim ↑ | BosDis ↑ | PosDis ↑ | S-PosDis ↑ | Disc. accuracy ↑ | Msg Variance ↓ |
|----------------|----------|----------|----------|------------|------------------|----------------|
| Reconstruction | 0.34 ±0.02 | 0.11 ±0.01 | 0.04 ±0.03 | 0.01 ±0.01 | 0.32 ±0.03 | 1334.38 ±78.05 |
| Discrimination | 0.09 ±0.01 | 0.32 ±0.19 | 0.19 ±0.13 | 0.01 ±0.00 | 0.62 ±0.05 | 2280.24 ±157.38 |

## E.2  Message purity over the Shapes dataset

Figure 9 presents the color purity, shape purity and max purity in each game type. See Appendix D.2 for their definitions. Table 5 shows the baseline values as well. The reconstruction-trained agents generate messages that convey more information about each attribute. However, the generated distribution of messages (which defines the random baseline) seems to have significant effect on the purity results. We hypothesize that higher multiplicity of attributes may better differentiate the two tasks.

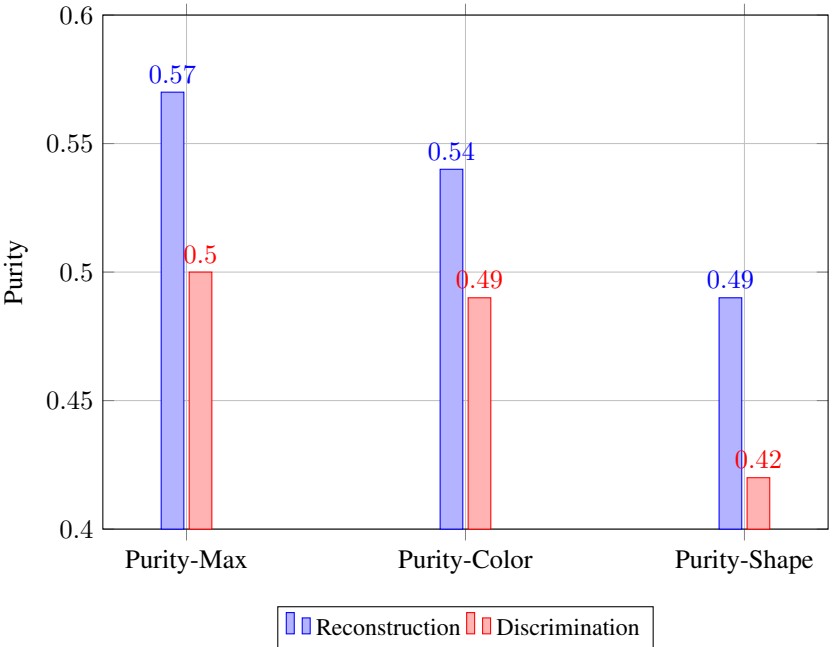

Figure 9: Message purity per attribute and game type.

Table 4: Correlation Matrices for All Data, Reconstruction, and Discrimination Groups

(a) Correlation Matrix (All Data)

|  | TopSim | BosDis | PosDis | S-PosDis | Accuracy | Msg Variance |
|---|---|---|---|---|---|---|
| TopSim | 1.00 | -0.63 | -0.63 | 0.49 | -0.97 | -0.97 |
| BosDis | -0.63 | 1.00 | 0.98 | -0.26 | 0.65 | 0.61 |
| PosDis | -0.63 | 0.98 | 1.00 | -0.29 | 0.61 | 0.62 |
| S-PosDis | 0.49 | -0.26 | -0.29 | 1.00 | -0.40 | -0.42 |
| Accuracy | -0.97 | 0.65 | 0.61 | -0.40 | 1.00 | 0.92 |
| Msg Variance | -0.97 | 0.61 | 0.62 | -0.42 | 0.92 | 1.00 |

(b) Correlation Matrix (Reconstruction)

|  | TopSim | BosDis | PosDis | S-PosDis | Accuracy | Msg Variance |
|---|---|---|---|---|---|---|
| TopSim | 1.00 | 0.33 | -0.37 | 0.76 | -0.24 | -0.61 |
| BosDis | 0.33 | 1.00 | 0.00 | 0.17 | -0.37 | -0.93 |
| PosDis | -0.37 | 0.00 | 1.00 | -0.02 | -0.61 | 0.26 |
| S-PosDis | 0.76 | 0.17 | -0.02 | 1.00 | -0.03 | -0.28 |
| Accuracy | -0.24 | -0.37 | -0.61 | -0.03 | 1.00 | 0.35 |
| Msg Variance | -0.61 | -0.93 | 0.26 | -0.28 | 0.35 | 1.00 |

(c) Correlation Matrix (Discrimination)

|  | TopSim | BosDis | PosDis | S-PosDis | Accuracy | Msg Variance |
|---|---|---|---|---|---|---|
| TopSim | 1.00 | 0.61 | 0.50 | 0.82 | 0.10 | 0.00 |
| BosDis | 0.61 | 1.00 | 0.99 | 0.07 | -0.04 | -0.23 |
| PosDis | 0.50 | 0.99 | 1.00 | -0.07 | -0.14 | -0.17 |
| S-PosDis | 0.82 | 0.07 | -0.07 | 1.00 | 0.26 | 0.12 |
| Accuracy | 0.10 | -0.04 | -0.14 | 0.26 | 1.00 | -0.80 |
| Msg Variance | 0.00 | -0.23 | -0.17 | 0.12 | -0.80 | 1.00 |

Table 5: Purity measures compared with random baselines.

|  | Purity-Max | | Purity-Color | | Purity-Shape | |
|---|---|---|---|---|---|---|
|  | Trained | Random | Trained | Random | Trained | Random |
| Reconstruction | 0.57 ±0.03 | 0.54 ±0.02 | 0.54 ±0.03 | 0.49 ±0.02 | 0.49 ±0.02 | 0.48 ±0.03 |
| Discrimination | 0.50 ±0.05 | 0.45 ±0.03 | 0.49 ±0.05 | 0.43 ±0.04 | 0.42 ±0.03 | 0.42 ±0.03 |

### E.3 Spatial meaningfulness empirical analysis

Recall that spatial meaningfulness requires close messages to have similar meanings. We propose an evaluation method that alters the message space such that it is well clustered, and evaluates the semantic consistency of message clusters rather than individual messages. To do so, we partition the 10 vocabulary symbols to five groups: $\{0, 1\}$, $\{2, 3\}$, $\{4, 5\}$, $\{6, 7\}$, and $\{8, 9\}$. During training, agents are restricted to using messages composed of symbols from a single group. This is implemented by letting the sender generate the first symbol freely, and then masking out all symbols except those belonging to the same set as the initial symbol for the remainder of the message generation. Consequently, messages such as $(3, 2, 3, 2)$ and $(7, 7, 7, 7)$ are possible, whereas a message like $(3, 4, 5, 9)$ can never be generated. There are 16 possible messages within each cluster, and only messages from the same cluster can have a Hamming distance smaller than 4 (the maximum). We then calculate spatial meaningfulness as the empirical unexplained variance (Algorithm 1) of these

clusters; a lower value indicates more spatially meaningful messages. Table 6 and Table 7 present this measurement over the validation sets of Shapes and MNIST respectively. While the reconstruction setting leads to a better improvement over the random baseline on Shapes, as expected by our findings, the results do not demonstrate high levels of spatial meaningfulness in any of the trained protocols. We hypothesize that the size of each cluster (16 messages) is too large, and perhaps a different partitioning method will provide more insight.

Table 6: Cluster Variance on Shapes.

| EC setup | Trained Cluster Var ↓ | Random Cluster Var | Improvement |
|---|---|---|---|
| Reconstruction | 3742.19 ±64.87 | 3915.19 ±70.52 | 4.41% |
| Discrimination | 3804.09 ±84.73 | 3919.11 ±73.73 | 2.94% |

Table 7: Cluster Variance on MNIST.

| EC setup | Trained Cluster Var ↓ | Random Cluster Var | Improvement |
|---|---|---|---|
| Reconstruction | 1790.88 ±29.88 | 1921.78 ±0.17 | 6.81% |
| Discrimination | 1745.86 ±29.47 | 1922.23 ±0.43 | 9.18% |
| Supervised disc. | 1783.60 ±44.15 | 1922.06 ±0.35 | 7.20% |

# F   Alternative formulations of the discrimination game

Recall that we formally model the discrimination receiver as a function $R\colon M \times \mathcal{X}^d \to \Delta\{1,\dots,d\}$ (Definition 2). In this setting, assuming an unrestricted hypothesis class, we show that the conditionally optimal receiver outputs a uniform distribution over the subset of candidates that belongs to the given message's equivalence class. This functionality is demonstrated in Figure 10.

In this section, we tackle two alternative formulations which provide stronger resemblance to common implementations of the discrimination game, at the cost of added complexity. The first alleviates the receiver's ability to consider more than one candidate at a time; the second explicitly models the prediction using dot product in a latent space. With the assumption of an unrestricted receiver hypothesis class, our results are persistent.

$$
\begin{aligned}
S(\mathsf{Candidate1}) &= m \\
S(\mathsf{Candidate2}) &\neq m \\
S(\mathsf{Candidate3}) &\neq m \\
S(\mathsf{Candidate4}) &= m \\
S(\mathsf{Candidate5}) &\neq m
\end{aligned}
\implies
R^*(m, \text{candidates}) =
\begin{cases}
\mathsf{Candidate1} & \text{w.p.} \quad 0.5 \\
\mathsf{Candidate2} & \text{w.p.} \quad 0 \\
\mathsf{Candidate3} & \text{w.p.} \quad 0 \\
\mathsf{Candidate4} & \text{w.p.} \quad 0.5 \\
\mathsf{Candidate5} & \text{w.p.} \quad 0
\end{cases}
$$

Figure 10: The conditionally optimal discrimination receiver, assuming it is available in $\Phi$.

**Candidate-unaware discrimination**

In this formulation, rather than directly outputting a distribution over the candidates, the receiver predicts a score for each of the candidates independently, and those scores are normalized to create the final probabilities. Referred to as the descriptive-only variation by [12]. Formally, the receiver takes the form

$$ R\colon M \times \mathcal{X} \to [0,1] $$

and the game objective is

$$
\ell(S, R, x) := \mathop{\mathbb{E}}_{\substack{t \sim U\{1,\dots,d\},\ x_t = x \\ x_1,\dots,x_{(t-1)}, x_{(t+1)},\dots,x_d \sim X^{d-1}}} - \log \frac{R(S(x_t), x_t)}{\sum_{i=1}^{d} R(S(x_t), x_i)}.
$$

The key difference is that each score only depends on the corresponding candidate and the message. However, the conditionally optimal receiver from Figure 10 can still be achieved, for example by the following receiver:

$$R^*(m, x) = \begin{cases} 1 & \text{if } S(x) = m \\ 0 & \text{otherwise} \end{cases}.$$

By assuming an unrestricted $\Phi$, this receiver is available and therefore optimal. Hence, the set of optimal communication protocols is the same in this setting as in the original one.

**Latent similarity discrimination**

This formulation further restricts the receiver by generating the scores via cosine similarity in a latent space, and using the softmax operator to create the final probabilities. This formulation closely resembles common implementations of the discrimination game. Formally, let $z \in \mathbb{N}$ be the latent dimension, the receiver now consists of two parts: an input encoder $R_X \colon \mathcal{X} \to \mathbb{R}^z$ and a message encoder $R_M \colon M \to \mathbb{R}^z$. The game objective becomes

$$\ell(S, R, x) := \mathop{\mathbb{E}}_{\substack{t \sim U\{1,\dots,d\}, \ x_t = x \\ x_1,\dots,x_{(t-1)},x_{(t+1)},\dots,x_d \sim X^{d-1}}} - \log \frac{\exp(\cos(R_M(S(x_t)), R_X(x_t)))}{\sum_{i=1}^d \exp(\cos(R_M(S(x_t)), R_X(x_i)))}.$$

Note that the softmax output is strictly positive in every entry, so the conditionally optimal receiver from Figure 10 cannot be achieved in this alternative formulation. However, we make the observation that the optimal input encoder outputs the same latent vector as the corresponding message encoder:

$$
\begin{aligned}
&- \log \frac{\exp(\cos(R_M(S(x_t)), R_X(x_t)))}{\sum_{i=1}^d \exp(\cos(R_M(S(x_t)), R_X(x_i)))} \\
&= - \log \left( 1 - \frac{\sum_{i \neq t} \exp(\cos(R_M(S(x_t)), R_X(x_i)))}{\exp(\cos(R_M(S(x_t)), R_X(x_t))) + \sum_{i \neq t} \exp(\cos(R_M(S(x_t)), R_X(x_i)))} \right) \\
&\geq - \log \left( 1 - \frac{\sum_{i \neq t} \exp(\cos(R_M(S(x_t)), R_X(x_i)))}{e + \sum_{i \neq t} \exp(\cos(R_M(S(x_t)), R_X(x_i)))} \right)
\end{aligned}
$$

and this lower bound can be achieved by selecting

$$R_X^*(x) = R_M(S(x)).$$

In other words, the conditionally optimal receiver needs to map equivalent inputs to the same latent vector (up to a multiplicative constant). In the case of unrestricted receiver hypothesis class, this still allows the relationship between equivalent inputs to be arbitrary. However, the representations of different messages are incentivized to be well separated, meaning that if similar inputs are mapped to different messages, their encodings by $R_X^*$ will be far apart. Restrictions on $R_X^*$ may render this infeasible, opening the way for semantic consistency in the discrimination setting. We leave further investigation of this idea to future work.

