# OpenReview forum: "Semantics and Spatiality of Emergent Communication"
_NeurIPS.cc/2024/Conference — NeurIPS 2024 poster_

### Official Review · Reviewer_o4TS · 2024-07-01

**Soundness:** 3
**Presentation:** 2
**Contribution:** 2
**Rating:** 7
**Confidence:** 4

**Summary:**

# Problem :

Emergent Communication (EC) protocols have been shown to be counterintuitive even when they enable agents wielding them to solve their related task.

# Contributions :

Rather than assuming that meaningful communication is taking place when a goal-oriented communication protocols enables the goal to be fulfilled by the agents wielding it, the paper identifies and formalises two goal-agnostic prerequisites to meaningful communication, the first one being coined **semantic consistency**, which relies solely on the claim that ‘inputs mapped to the same message should be semantically similar as is the case with human language’, and the second one entitled **spatial meaningfulness**, which is stricter for ‘it also accounts for distances between messages’ in the language space.

The paper then proceeds to analyse the two common EC environments (reconstruction vs discrimination settings) with those concepts, showing both theoretically and experimentally (with the MNIST dataset) that:

1. in the reconstruction setting, every optimal solution is semantically consistent ;

2. providing (i) a 'bounding [of] the rate of change in receiver's output' is applied and (ii) the receiver does not degenerate, then optimal reconstruction-setting solutions are also spatially meaningful, which is opposed to …

3. … the fact that optimal discrimination-setting solutions need not be semantically consistent nor spatially meaningful.

The later (3) is proven thanks to the interesting insight (in Corollary 5.4) that a uniformly random communication protocol can be a globally optimal _discrimination-setting_ solution, thus meaning that its input-to-message mapping can be arbitrary, i.e. meaningless.

**Strengths:**

## Originality :

To my knowledge, the contributions made here on a theoretical standpoint are novel and valuable.

## Quality :

SQ1: Acknowledgement of the ‘global discrimination’ context (in footnote 1) and discussion around the difficulty to measure the quality of the receiver’s reconstruction is very insightful. It might be worth expending and/or emphasising further even (possible citation of [1]’s section 2.7 parag. 2, where the Task & Talk game [2,8] is highlighted as an instance of it), by bringing the nuance made in ln446-447 back into the main text, for instance.

SQ2: Section 2.3 is very insightful and provides a lot of important connections to understand the current work and its stakes.

SQ3: The definition of an EC setup as a sextuple that allows representing both discrimination and reconstruction settings in a common framework is very insightful and a worthwhile contribution to the field, I think.

## Clarity :

SC1: Well-written introduction that provides context and stakes with clarity.

## Significance :

SS1: I think the EC setup (cf. SQ2) is of great significance to the EmeCom field.

SS2: I think the semantic consistency is a valuable theoretical contribution, but it is unclear how it relates to other aspects in practice and therefore I would hope for the authors to revise the paper towards providing more significance for it (cf. weaknesses below with some proposal improvements).

**Weaknesses:**

# Weaknesses :

## Originality :

WO1: Missing citation in Section 2.2 of [15]

## Quality :

WQ1: possible missing citations regarding the introduction of choices for the discrimination task [1: section 2.6.1. highlighting [3]] and the impact on the resulting protocols [3: section 3.7].

WQ2: ln 92-94 : missing citations for the definition of compositionality, e.g.  “the meaning of a complex expression is a function of the meaning of its immediate syntactic parts and the way in which they are combined” [4] ;

ln94-95 : possible missing citations regarding the argument that compositionality does not correlate with generalisation: [5] and [6], especially in visual domain (as opposed to [paper 7] being constrained to symbolic domain).

WQ3: following SQ2, I would be inclined to argue for the latter end of section 2.3 to be further detailed in order to make it more accessible to a wider audience, possibly by defining MI in the context of EmeCom already here, maybe? My point is that the discussion is citing a lot of papers (which is great for attribution and significance), but it does not help the not-specialist reader to understand the stakes concretely. A possible solution could be to separate background and related works into two sections and push the related works section at the end of the paper, in order to allow it to reference back at the important concepts within the paper, with the benefit of the reader having hopefully acquired some familiarity with the work at large by the time they reach this section.

WQ4: ln220 after Lemma 5.1 provides some valuable insights, especially with respect to the ‘unexplained variance’, but the statement feels too shallow without a concrete equation defining what is ‘unexplained variance’ for instance. I would like to urge the author to make it more concrete in order to increase the quality of that insight and the corresponding lemma.

## Clarity :

WC1: meaningfulness and ‘similarity to natural language’ are often talked about together (e.g. 44-45, 50-51), without having been clearly related to each other before. I understand that it may seem obvious that the adjective ‘meaningful’ means ‘meaningful to human beings or from the comparison with human languages’ and therefore a ‘meaningful’ emergent language should have high similarity with human/natural languages, but the general audience might appreciate a warning on that association. It also offers an opportunity for the paper to address concepts and related work around (human) alignment, which I assume would increase its significance. This is a rabbit hole that can go as far as the work of [10] (and [11] that instantiates it in the context of EmeCom).

WC2: ln178-179 : the paper states that in EmeCom papers 'the discrete message space is usually much smaller than the input space', which I think is a misrepresentation and it also loses an important nuance regarding the impact of the communication channel's capacity, as studied in [5,8,9] for instance, where [8,9] finding that a ‘restrictive’ communication channel  is a driver of compositionality in the emergent language, when using a REINFORCE-like algorithms with symbolic stimuli, in contrast to [5] finding the opposite when using the (Straight-Through) Gumbel-Softmax approach with visual stimuli: overcomplete communication channel (where the max sentence length is increased) yields higher compositionality than complete ones. This latter is very relevant to the paper given its usage of Gumbel-Softmax approach and visual stimuli (MNIST).

My point is that this sentence in the paper occludes and misrepresents an important inquiry in EmeCom, and I would urge the authors to reformulate appropriately.

WC3: the term 'unrestricted' is defined in ln148 and ln157, but I fail to understand what it means exactly in this context and when is its opposite considered ?

WC4: In Appendix C.2 Lemma proof, the derivations below ln533 are ambiguous to me, especially the third line. I would like to invite the authors to make all of their proofs easier to understand by explaining what property or equation is used to reach each derivation. My point is that since there is no space restriction in the appendices, the paper would be made easier to understand by putting as many explanations as possible. I would hope for us to break away with this legacy/bias of taking ‘mathematical derivations being difficult to read’ as a badge of quality or importance of the derivations presented (assuming that this is the implicit bias that was at play here).

WC4: Theorem 5.2 requires $\Theta$ to contain ‘at least one semantically consistent protocol’. This requirement is surprising to me. It would increase the clarity of the paper if some insights or motivation for it could be provided, please?

WC5: Acronym ‘KKT’ used in ln567-568 without definition.

WC6: Equations could be more easily reference with numbering in all cases.

## Significance :

WS1: missing citation for the 'classification discrimination' setting (and a lesser extent the 'supervised discrimination' context) defined in ln169-172 to [1] that defines it as an object-centric (as opposed to stimulus-centric) and descriptive variant, highlighting it is first instantiated by [7]. It is later investigated in [6]. Adding those references would show the reader that those ideas have some precedence and are of value to the field of EmeCom, and provide adequate attribution.

WS2: Experimenting with MNIST is not motivated in the paper, and not really in line with previous works in EmeCom, for instance because (i) it does not have much structure, and (ii) it does not have annotations of different attributes and values that would allow compositionality(/disentanglement) metrics to be computed, which is of concerned here.

Thus, I would hope for the paper to propose some experiments with symbolic stimuli for starters, following the work of [paper 7], or with a (Colored) dSprites dataset [13 - similarly to [14]] or 3dshapes dataset [12 - visual dataset with a lot of structure, i.e. up to 6 attributes], or the simplified dataset from [7, reproduced in an open-source version in 6].

Moreover, in order to evaluate systematic generalisation and how it relates to semantic consistency, the experiments should use a train/test split strategy as shown in [paper 7] for symbolic stimuli, or [5 Sec. 3, 6] to be more in touch with dSprites and 3dshapes-like visual datasets.

WS3:  the paper does not provide a clear-cut metric for semantic consistency despite the clear definition in Definition 3 (ln195). ln302 does mention using message variance and points to Appendix E.2, but I think the paper would be made more impactful and clearer if this had been discussed around Definition 3 too and made very explicit. The same goes for spatial meaningfulness (ln307) despite definition 4 in ln257 not mentioning the related metric in a clear-cut fashion. My point is that I think that the paper would be more significant by providing very clear metrics that the community can adopt. Obviously, open-sourcing the codebase to compute those metrics would go a long way in that direction.

WS4: Following up on WS3, in order to build a bridge between the previous works on compositionality in EmeCom **as a proxy for human-like communication/naturalness** (as acknoeledge in Section 2.2), and the current paper that proposes to evaluate human-like communication using the proposed definition of semantic consistency, I would have expected the paper to report some correlation metrics between relevant compositionality metrics (topsim, and posdis [paper 7], at least, and possibly the refined speaker-centred posdis from [6] at best) and a proposed semantical consistency metric (e.g. explained variance). These correlation measures would enable the community to reflect on the extent to which the compositionality-as-proxy-for-naturalness approach is lacking in terms of semantics and spatiality.

My point is that the current contribution is interesting and its qualitatively value has been well-discussed, but it is unclear how it relates to the previous works in a quantitative way. For instance, when considering quantitative results, are semantic consistency and compositionality just two sides of the same coin or do they capture really different aspects of human-like communication?

WS5: Following up on WS2 and WS4, another valuable contribution that would allow comparison with previous works in EmeCom could be to evaluate the systematic generalisation of the resulting communication protocol and measure their correlation with a semantic consistency metric in order to clarify the extent to which semantic consistency aligns with systematic generalisation, which has been of great interest in EmeCom.

This would increase the significance of the work presented here, but it might be slightly out of the scope. In any case, I would like to inquire on the author’s perspective: why not having conssidered it so far? What kind of correlations would you expect? Can the formal setup provided in the paper and the concept of optimality it relies on say anything about systematic generalisation, please?

**Questions:**

Please see within the strenghts and weaknesses, but mainly the following:
1) Please see WC3.

2) Please see WC4.

3) Please see WS3 and WS4.

4) Please see WS5.


# References:

[paper 7]: Chaabouni, Rahma, et al. "Compositionality and Generalization in Emergent Languages." ACL 2020-8th annual meeting of the Association for Computational Linguistics. 2020.

[1] : Denamganaï, Kevin, and James Alfred Walker. "ReferentialGym: A Nomenclature and Framework for Language Emergence & Grounding in (Visual) Referential Games." arXiv preprint arXiv:2012.09486 (2020), 4th EmeCom Workshop @ NeurIPS 2020.

[2]: Cogswell, Michael, et al. "Emergence of compositional language with deep generational transmission." arXiv preprint arXiv:1904.09067 (2019).

[3]: Lazaridou, Angeliki, et al. "Emergence of Linguistic Communication from Referential Games with Symbolic and Pixel Input." International Conference on Learning Representations. 2018.

[4]: M. Krifka. Compositionality. The MIT encyclopedia of the cognitive sciences, pages 152–153, 2001.

[5]: Denamganaï, Kevin, and James Alfred Walker. "On (emergent) systematic generalisation and compositionality in visual referential games with straight-through gumbel-softmax estimator." arXiv preprint arXiv:2012.10776 (2020). 4th EmeCom Workshop @ NeurIPS 2020.

[6]: Denamganaï, Kevin, Sondess Missaoui, and James Alfred Walker. "Visual Referential Games Further the Emergence of Disentangled Representations." arXiv preprint arXiv:2304.14511 (2023).

[7]: Choi, Edward, Angeliki Lazaridou, and Nando de Freitas. "Compositional Obverter Communication Learning from Raw Visual Input." International Conference on Learning Representations. 2018.

[8]: Kottur, Satwik, et al. "Natural Language Does Not Emerge ‘Naturally’in Multi-Agent Dialog." Proceedings of the 2017 Conference on Empirical Methods in Natural Language Processing. 2017.

[9]: Resnick, Cinjon, et al. "Capacity, Bandwidth, and Compositionality in Emergent Language Learning." Proceedings of the 19th International Conference on Autonomous Agents and MultiAgent Systems. 2020.

[10]: Santoro, Adam, et al. "Symbolic behaviour in artificial intelligence." arXiv preprint arXiv:2102.03406 (2021).

[11]: Denamganaï, Kevin, Sondess Missaoui, and James Alfred Walker. "Meta-Referential Games to Learn Compositional Learning Behaviours." arXiv preprint arXiv:2207.08012 (2022).

[12]: [https://github.com/google-deepmind/3d-shapes](https://github.com/google-deepmind/3d-shapes)

[13]: [https://github.com/google-deepmind/dsprites-dataset](https://github.com/google-deepmind/dsprites-dataset)

[14]: Xu, Zhenlin, Marc Niethammer, and Colin A. Raffel. "Compositional generalization in unsupervised compositional representation learning: A study on disentanglement and emergent language." Advances in Neural Information Processing Systems 35 (2022): 25074-25087.

[15]: Brandizzi, Nicolo. "Toward More Human-Like AI Communication: A Review of Emergent Communication Research." IEEE Access 11 (2023): 142317-142340.

**Limitations:**

## Limitations & General advice:

I think the paper’s experiment section is the main issue of the paper for it is too shallow and not performed on a relevant dataset.
By shallow, I mean that it fails to provide insights with respect to previous works, which limits its overall impact onto the community.
I have highlighted in the weaknesses related to significance how this could be addressed, as far as I am concerned.

Overall, my appreciations of the paper is currently only at 5 because of those limited evaluation, but if this can be addressed then I would raise my overall rating to 8, and my contribution rating to excellent.

It will be necessary to find some space to include some of those considerations.
Thus, I would advise the authors to replace most of their \subsection with \textbf{Subsection Title.} without line jump, for it would gain about 3 lines for each subsection titles, without hurting clarity.

Figure 3 is great for insights but it occupies a lot more space than necessary, I think. Moving it all into a vertical \wrapfigure should be helpful to gain space, without impairing clarity. Subsequently, moving the subcaptions into the main caption and referring to each part as \textbf{Top/Middle/Bottom} would be helpful in gaining some space too.

ln203-208’s itemize could be replaced with normal text and ‘(i)’ and ‘(ii)’ before each contributions, for instance, to gain 3 lines of content probably. Similar treatment for ln261-266’s itemized can be applied.

Lemma 5.1’s equation could enable one extra line by removing the assumpted line jump.

Ln 252 and the associated replication of Equation 1 could be spared by simply referring to equation 1, which would gain about 3 lines of space..

# POST REBUTTAL UPDATE :

Most of my concerns have been adressed by the rebuttals, with the exception of some clarifications around WS2/WS4 and some possible further revisions regarding WS5, therefore I am increasing my overal score to 7.

---

> ### Author Rebuttal · Authors · 2024-08-06
>
> Thank you for this insightful and detailed review. We are glad you have found our theoretical contributions novel and valuable. The papers that you have mentioned, especially regarding compositionality and input representation, are indeed relevant to our core contributions, and we will add references to them in our next revision.
>
> We will now address your comments individually:
>
> SS2: We agree that comparing our definitions to existing ideas is important. An interesting connection between our definitions and Topographic Similarity (topsim) is that while topsim considers the relationship between every pair of inputs, our definitions only consider pairs that correspond to similar messages. The latter follows an intuitive asymmetry: inputs with similar messages are expected to be similar, but inputs with dissimilar messages do not have to be different. Further discussion will be added in the next revision.
> On the empirical front, new experiments are described in the response for comment WS2, including correlation with topsim, posdis, bosdis, and speaker-posdis.
>
> WO1, WQ1, WQ2: The missing citations will be added in the next revision. Thank you for the references.
>
> WQ3: Section 2.3 will be expanded in the next revision.
>
> WQ4: This explicit equation for explained variance will be added to the relevant paragraph: $\text{Var}_{m \sim S(X)}\mathbb{E}[X | S(X)=m]$
>
> WC1: This is a good observation, which aligns with a central theme in our paper: distinguishing "solves the task" from "has (consistent) meaning." In the next revision, we will clarify that "meaningful" refers to human-recognizable properties.
>
> WC2: We apologize for the mischaracterization. Our framework is mainly applicable to setups with a small-capacity messaging channel, where inputs necessarily get mapped to the same message. Nonetheless, research investigating larger message spaces is an important part of EC literature, including alternative types of messages such as quantization or explicit codebooks. We will reformulate the relevant paragraph accordingly.
>
> WC3: The term 'unrestricted' means that the agent can assume any function from its domain to its range. This assumption allows closed-form, interpretable solutions for synchronized receivers. However, it is unrealistic in the context of natural language, as physical and mental limitations play an important role in shaping human communication. Removing this assumption motivated the second part of our paper (dealing with spatial meaningfulness), where agents are not assumed to be unrestricted.
>
> WC4(1): We apologize for any confusion and definitely did not intend for the proofs to be hard to read. The referred-to third line is the law of total expectation, where the events are the assignment to message m. We have made efforts to ensure all proofs are written clearly and rigorously, so their correctness is easy to assert. We will add additional explanations for derivations in our revision.
>
> WC4(2): This assumption simply prevents the degenerate case where every available communication protocol is semantically inconsistent, which would render every optimal solution inconsistent regardless of the objective function.
>
> WC5: The KKT conditions characterize optimal solutions to convex problems. We will add the appropriate citation [2].
>
> WC6: Thank you for the suggestion, we will add equation numbering to our lemmas and to key equations throughout the proofs section for easier reference.
>
> WS1: The missing citations will be added in the next revision. Thank you for the references.
>
> WS2: We fully agree with your comment. The empirical analysis will better support our findings if we experiment with more sophisticated data. With this in mind, we have conducted additional experiments on a special version of the Shapes dataset [1] where each input is an image of an object with several sampled attributes (shape, color, position, rotation, and size). The multiplicity of attributes allowed us to calculate compositionality/disentanglement measures from the literature, including your suggestions. The results are appended to the general author rebuttal (and will be added to the paper in the next revision).
>
> The results show a significant gap in semantic consistency (lower message variance is preferable), on par with the previous evaluation on MNIST. Additionally, topographic similarity strongly favors the reconstruction setting (0.37, versus 0.08 for discrimination), confirming a positive relationship between semantic consistency (as measured by message variance) and compositionality (as measured by topsim). We also report correlations between every pair of measures, taken over 10 runs (5 for each task). These correlations should be considered with caution as they are affected by the inclusion of both tasks for calculation. We may report separate tables for each game type to avoid this issue.
>
> WS3: We prefer to keep the theoretical and empirical parts separate to avoid splitting attention. However, it is crucial to present the metrics before the results, so we restructured the empirical section to mention the metrics before introducing any results. As for the codebase, it will be open-sourced with the publication, and you can access it now via the zip attached to the original submission.
>
> WS4: See our responses to WS2 and SS2.
>
> WS5: Systematic generalization is not included in our theoretical framework. Given our recent results, we might expect semantically consistent protocols to generalize better to unseen attribute values, as the relationship between topsim and systematic generalization has sometimes been shown to be positive.
>
> We greatly appreciate your space-saving tips. We will try to implement them in the next revision as well.
>
> [1] Kuhnle, Alexander and Copestake, Ann. "ShapeWorld - A new test methodology for multimodal language understanding". arXiv preprint arXiv:1704.04517
>
> [2] Bertsekas, Dimitri P. "Nonlinear programming". Journal of the Operational Research Society.

---

> > ### Comment · Reviewer_o4TS · 2024-08-13
> > **Reply**
> >
> > Thank you for your answers on each of my review points. I am adressing below the remaining concern:
> >
> > ## WS2/WS4 :
> > Thank you for your care in addressing this concern, I find that it greatly improves the quality of the contribution.
> > Nevertheless, I have some questions:
> > 1. Could you clarify how you measured Discr. accuracy in the context of the Reconstruction task, please?
> > 2. Are you reporting measures all measures over training or testing data splits or over the whole dataset, please? (c.f. [5] for details about the possible issue: namely reporting over test data split alone is more insightful)
> > 3. Are you performing a uniform train/test split or a zero-shot compositional train/test split, please? (c.f. [1,5,6])
> >
> > ## WS5 :
> > I understand that systematicity is not part of your theoretical framework, but your paper is placing itself in the EmeCom literature for which systematicity is an important concern.
> > Depending on your answer to question 3 above, if you are performing zero-shot compositional train/test splits, then I think that there is only little work needed for your experimental results to provide contributions on the systematicity front, even if only restricted to experimental contributions.
> >
> >
> > ## Score Update:
> > Based on the answers and what has been shown to be feasible so far in terms of revision, I am increasing my overall score to 7 and reserve the possibility of raising it further depending on the current discussion points.

---

> ### Author Response · Authors · 2024-08-13
> **Reply**
>
> Thank you for your response and for adjusting the score. We appreciate your feedback.
>
> ### **Discrimination accuracy in the reconstruction setting**
>
> We define the referential prediction of the reconstruction setup as the candidate that minimizes the loss when paired with the target's reconstruction. Namely, we pass the given message to the receiver agent which outputs a reconstruction of the target, and then pick the candidate with the lowest Euclidean distance to that output.
>
> ### **Train/test split**
>
> The results are measured over the test split only. The train/test split is done uniformly, without splitting values of specific attributes.
>
> ### **Systematic generalization**
>
> We wish to emphasize that the primary contributions of this paper are theoretical, and that the empirical analysis is intended to support those findings. The relationship between our definitions/measures and systematic generalization is interesting and worth exploring, but we prefer to keep the experiments within the scope of the existing theoretical framework. We do recognize, however, that future works could offer significant value to the field by extending both the theory and experiments to explore systematicity.
>
> If you would like any further clarifications or have additional concerns, please let us know.

---

### Official Review · Reviewer_TFDu · 2024-07-13

**Soundness:** 3
**Presentation:** 3
**Contribution:** 2
**Rating:** 6
**Confidence:** 3

**Summary:**

The authors consider a collaborative multi-agent (2 agent) setting with inter-agent communication, where the communication protocol is learned by the agents in order to maximize their common objective, which can be either a reconstruction or discrimination task. However, the authors note that this communication protocol that maximizes their common objective need not be meaningful. The authors define meaningful communications as those that have semantic consistency across different objectives or tasks. The authors show that under some technical conditions that semantically consistent protocols optimize discrimination tasks but not reconstruction tasks. They also show that reconstruction tasks require spatial meaningfulness as well, which is defined as a stricter property than semantic consistency. Also, the authors show that all optimal solutions to the reconstruction task are semantically consistent and also spatially meaningful (under some different assumptions), however semantic consistency is not a necessary condition for optimal protocols for discrimination tasks and neither is spatial meaningfulness. They validate these theoretical results with some numerical experiments using MNIST.

**Strengths:**

The paper is very well written and all the theorems and proofs are clearly written. The problem is interesting and the theoretical tools presented by the authors will be useful in several other problems as well.

**Weaknesses:**

The authors define two properties for meaningfulness of messages, but in their empirical evaluation, they are unable to show the importance of their more restrictive property - spatial meaningfulness.

**Questions:**

1. Since the setting considered is a collaborative multi-agent system, why don't the agents communicate the raw information that they have to each other? Is this due to communication channel or cost related limitations? A related question: what leads to the "discrete bottleneck" mentioned in line 45?
2. Why are only reconstruction and discrimination tasks considered? What communication protocols emerge in multi-agent games such as football (for example, multi-agent communication in https://github.com/LARG/HFO?tab=readme-ov-file , which is a simulated soccer environment) ?
3. What does "...protocols created via the discrimination task often generalize extremely well to random noise data,..." in Line 42 mean? Is it that such protocols allow a receiver to accurately classify a random noise input or does it imply that the receiver is robust to noise? Why is this property compared with "human-recognizable properties",  since human communication is not usually fully collaborative and the objectives are also more complex than discrimination and reconstruction?
4. Why is similarity to natural language a desired property for such inter-agent communication protocols? Is it for human-in-the-loop scenarios or for debugging the system by a human?
5. Shouldn't the EC setup objective given immediately after Line 142 also include $M$, the message space as a conditioning variable?
6. With the statement: "Note that a pair of agents can be synchronized in both directions without being optimal.", do the authors imply that constraining the $\theta$ and $\phi$ space might lead to sub-optimal protocols or unilateral (independent or disjoint) optimization by the sender and receiver will lead to sub-optimal protocols?
7. Is the communication channel assumed to be noiseless?
8. Have the authors considered comparison with Vector Quantized auto-encoders as well?
9. The additional input given in the Discrimination task is not shown in Figure 1. Is there a reason for this?
10. How are the independently sampled distractors communicated to the receiver agent in the discrimination task? Don't channel limitations apply here?
11. In Figure 3, are the colours and shapes indicative of the input class or are they (partial) input properties? If so, can this be added to the caption?
12. The equation below Line 213 is not clear. The arg min operator is set-valued, so $\in$ is more appropriate than $=$.
13. Does the synchronized receiver setup lead to some loss of optimality?
14. Can the authors define the "Binomial" function and their notation in Lemma 5.3?
15. Can the condition given in Definition 5 be ensured/verified apriori, in practice?
16. What is the motivation/intuition behind Definition 6? Specifically why is the $\frac{1}{4}$ term chosen?
17. In Page 14, what is $i$ in $S(i)$ the third line of the last set of equations? Shouldn't this be $S(x)$?
18. What are the differences in the proof of Lemma 5.1 with reference [33]?
19. Typo: Line 541, "for" is not needed here.
20. Can the $\cdot$ (dot) symbol imply in the equation in last line on Page 15 be defined and brackets added here to make reading this expression easier?

**Limitations:**

The authors adequately address the limitations in a separate section in the main paper itself.

---

> ### Author Rebuttal · Authors · 2024-08-06
>
> Thank you for this insightful review. We are glad you have found the novel theoretical tools useful. Following your feedback and others, we have prepared some major modifications to be added in the next revision.
>
> You have mentioned as a weakness the empirical results with regard to spatial meaningfulness (the "cluster variance" metric"). We would like to emphasize that the key contributions of our work are theoretical, and that the empirical analysis is intended to be a complementary proof-of-concept supporting our findings. With that said, we admit that our proposed metric for spatial meaningfulness does not yield interesting results, unlike its counterpart (message variance). We hypothesize that the size of each cluster (16 messages) is too large, and perhaps a different partitioning method will provide more insight. This hypothesis will be added in the next revision.
>
> We now like to address your questions.
>
> 1. The agents cannot communicate raw information due to the limited capacity of the communication channel, i.e., the "discrete bottleneck". In most EC setups, the message space is constructed as sequences of tokens and is therefore often small in comparison to the input space. For example, in our MNIST experiments, there are 10K possible messages, which is smaller than the size of the dataset.
> 2. There are several benefits of simple two-agent communication games. To name a few: optimization is easier, language analysis is more straightforward (there is only one speaker), and each message functions as a latent representation of the input, so concepts from representation learning literature can be applied to study the setup.
> With that said, EC with many agents is also studied frequently, but outside the scope of this paper.
> 3. The experiment that revealed the agents' ability to perform well on random noise [1] trained discrimination agents on natural images (imagenet) and then performed inference where both the target and the (single) distractor are just sampled noise. Surprisingly, the agent were able to play the game (i.e., the receiver was able to detect the target) with high accuracy. This result indicates that the features perceived by the agents are not high-level classes or objects ("human-recognizable properties"), illustrating the potential for counterintuitive communication protocols to emerge.
> 4. The goal of inducing characteristics of natural language in communication protocols is what often sets apart EC from related fields such as Multi-Agent Reinforcement Learning (MARL), which focus on long-term expected reward. Creating learning environments that induce protocols similar to natural language is desirable in several aspects: it may improve our understanding of the underlying deep learning architectures, advance the goal of human-AI interaction, and even provide insights into language evolution.
> 5. It doesn't have to, because M is embedded into the functions S and R.
> 6. A situation where both agents are synchronized is a situation where neither agent can 'do any better' given the fixed state of the other (Nash equilibrium). This pair of agents does not necessarily minimize the expected loss. This kind of suboptimal equilibrium can happen with an alternating optimization method (e.g., local minimum at the end of kmeans).
> 7. Yes. Investigating the effect of noise in the communication channel is an interesting idea for future work.
> 8. Quantization is a viable discretization method for EC research. However, it often defines very large message spaces, which means that most used messages likely represent a single input. For that reason, it does not fit the required setting for our analysis.
> 9. Sorry for the confusion. We avoided additional arrows to maintain an elegant figure. To clarify this, we added an explanation in text below the figure.
> 10. The distractors are given to the receiver by the game, as independent input. The only thing going through the communication channel is a single message, which the sender generates based on seeing only the target (the "correct" candidate).
> 11. Right, this is an important clarification. The input itself is only the location of each object (a vector), not the attributes themselves. We used shapes and colors to illustrate how distance between inputs can be indicative of their semantic similarity. This clarification has been added to the caption for the next revision.
> 12. Thank you for the observation, we fixed it.
> 13. It does not. By definition, a synchronized receiver minimizes the loss for a specific sender. As a result, when we plug in the closed-form synchronized receiver into the loss function, we get the best achievable loss value for the sender (as a function of the sender agent).
> 14. We have added a clarification note, the word "Binomial" refers to the binomial distribution, over which the expectation is taken.
> 15. We are not sure how one should go about verifying this assumption in practice. While the constant k can be empirically estimated, one would have to bound (from above) the norm of the receiver's gradient to ensure that this definition is satisfied.
> 16. The motivation behind the non-degeneracy definition is to prevent a situation where the fixed receiver in theorem 6.1 is constant, which means that the sender agent doesn't affect the game performance, and therefore isn't incentivized to use a spatially meaningful communication protocol. Our proof for theorem 6.1 requires the receiver to be strictly better than constant and this bound is performed twice, so any constant smaller than ½ can replace the ¼ in definition 6.
> 17. Indeed a typo. Thank you for the observation, we fixed it.
> 18. We use slightly different settings and notations, so we chose to include the proof for completeness.
> 19. Thank you for the observation, we fixed it.
> 20. Of course. Done.
>
> [1] Bouchacourt, Diane and Baroni, Marco. "How agents see things: On visual representations in an emergent language game". arXiv preprint arXiv:1808.10696

---

> > ### Comment · Reviewer_TFDu · 2024-08-13
> >
> > I thank the authors for their detailed responses to my questions. Based on reading these and the other reviews as well, I would like to retain my score.

---

### Official Review · Reviewer_XigD · 2024-07-14

**Soundness:** 3
**Presentation:** 2
**Contribution:** 2
**Rating:** 5
**Confidence:** 4

**Summary:**

This paper explores the properties of communication protocols that emerge when artificial agents are trained to perform collaborative tasks through a communication channel. The authors identify a key prerequisite for meaningful communication, termed "semantic consistency," which demands that messages with similar meanings should be used across different instances. They provide a formal definition for semantic consistency and use it to compare two common objectives in emergent communication research: discrimination and reconstruction. The paper proves that under certain assumptions, semantically inconsistent communication protocols can be optimal for discrimination tasks but not for reconstruction. It introduces a stricter property called "spatial meaningfulness," which considers the distance between messages and aligns more closely with natural language characteristics. Experiments with emergent communication games validate the theoretical findings, showing an inherent advantage in communication goals based on distance rather than probability.

**Strengths:**

- The formal definitions and theoretical proofs are innovative and contribute to the literature by setting new standards for what constitutes meaningful communication in EC systems.
- The paper is well-structured, with clear definitions, hypotheses, and theorems that build logically upon one another.

**Weaknesses:**

- The empirical validation using the MNIST dataset is a good start but may not fully capture the complexity of real-world applications. The paper could benefit from additional experiments with more diverse datasets or more complex tasks to further validate the theoretical findings.
- The paper introduces the concept of spatial meaningfulness, which considers distances between messages. However, the empirical analysis of spatial meaningfulness using message clusters may not fully explore the nuances of this concept. Further investigation into different methods of evaluating spatial meaningfulness could provide more actionable insights.
- The author assume that Euclidean distance broadly indicates the level of semantic similarity. However, depending on the context, other distance metrics might be more appropriate. The paper could be improved by discussing the limitations of using Euclidean distance and the potential impact of using alternative metrics.

**Questions:**

See weakness

**Limitations:**

- The paper mentions a two-stage training procedure involving a continuous autoencoder, but it does not delve into the computational efficiency of the proposed methods. Including a discussion on the scalability of the approach and its computational requirements would provide a more comprehensive understanding of its practical applicability.

---

> ### Author Rebuttal · Authors · 2024-08-06
>
> Thank you for this insightful review. We are glad you have found the theoretical part innovative and well structured. Following your feedback and others, we have prepared some major modifications to be added in the next revision. We would now like to address your comments one by one.
>
>
> **Comment 1: The paper could benefit from additional experiments with more diverse datasets or more complex tasks to further validate the theoretical findings.**
>
> Indeed. While our key contributions are theoretical, we agree more experiments can help validate our findings. While the simplicity of MNIST has its benefits, we have recently conducted additional experiments on a special version of the Shapes dataset [1], where each input is an image of an object with several sampled attributes (shape, color, position, rotation, and size). The multiplicity of attributes allowed us to calculate compositionality/disentanglement measures from the literature, and the results are appended to the general author rebuttal (and will be added to the paper in the next revision).
>
> The results show a significant gap in semantic consistency (lower message variance is preferable), on par with the previous evaluation on MNIST. Additionally, topographic similarity strongly favors the reconstruction setting (0.37, versus 0.08 for discrimination), confirming a positive relationship between semantic consistency (as measured by message variance) and compositionality (as measured by topsim). We also report correlations between every pair of measures, taken over 10 runs (5 for each task). These correlations should be considered with caution as they are affected by the inclusion of both tasks for calculation. We may report separate tables for each game type to avoid this issue.
>
> **Comment 2: The empirical analysis of spatial meaningfulness using message clusters may not fully explore the nuances of this concept. Further investigation into different methods of evaluating spatial meaningfulness could provide more actionable insights.**
>
> Investigating other estimation methods of spatial meaningfulness would indeed be an interesting addition. We wish to emphasize that the major contributions of this paper are theoretical, and the empirical analysis is intended to be a complementary proof-of-concept supporting our findings. With that said, we admit that our proposed metric (cluster variance) for spatial meaningfulness does not yield interesting results, unlike its counterpart (message variance). We hypothesize that the size of each cluster (16 messages) is too large, and perhaps a different partitioning method will provide more insight. This hypothesis will be added in the next revision.
>
> **Comment 3: The paper could be improved by discussing the limitations of using Euclidean distance and the potential impact of using alternative metrics.**
>
> We do mention this limitation on page 6 (footnote) and in the limitations section, but perhaps not elaborately enough. In some types of data, like images, Euclidean distance on raw pixels might not be indicative of desirable properties. Note that all our theoretical results hold anyway, and that some of our results do not depend on distance (e.g., Corollary 5.4). Furthermore, a solution that we implement in our experiments is the use of embeddings generated by a pretrained continuous model. Within that embedding space, the meaningfulness of Euclidean distance becomes an easier assumption. This discussion will also be added to the next revision.
>
> **Comment 4: Including a discussion on the scalability of the approach and its computational requirements would provide a more comprehensive understanding of its practical applicability**
>
> Appendix D is written precisely to supply technical information regarding the experiments. The first stage of training is a continuous autoencoder training procedure, which is very quick (especially on MNIST). It takes approximately 10 minutes using a standard GPU. This detail will be included in the next revision.
>
> [1] Kuhnle, Alexander and Copestake, Ann. "ShapeWorld - A new test methodology for multimodal language understanding". arXiv preprint arXiv:1704.04517 (2017)

---

> > ### Comment · Reviewer_XigD · 2024-08-14
> >
> > After reading the rebuttal of the authors and the discussion between the authors and other reviewers, I think my concerns are mainly addressed. I would raise my rating.

---

> > > ### Author Response · Authors · 2024-08-14
> > > **Reply**
> > >
> > > Thank you for adjusting the score. We appreciate your feedback and would love to address any remaining issues.

---

### Official Review · Reviewer_cajw · 2024-07-30

**Soundness:** 2
**Presentation:** 3
**Contribution:** 2
**Rating:** 3
**Confidence:** 5

**Summary:**

This paper investigates and analyzes the emergent communication protocols developed by agents during collaborative tasks that necessitate message transmission to solve given problems. The authors contend that traditional performance measures, including task performance and properties like compositionality and efficiency, are insufficient for ensuring that emergent communication is meaningful or interpretable, especially in the context of human communication evolution. To address this gap, they introduce the concepts of semantic consistency and spatial meaningfulness, inspired by observations of human communication and the properties of emergent communication (EC) protocols influenced by the discrete bottleneck. The authors propose theoretical solutions to assess the similarities of emergent protocols to natural language and conduct empirical tests to support their theory. Their study revolves around an EC setup inspired by Lewis’ games, where a sender agent describes a given input, and a receiver agent makes predictions based on that description. The authors focus on reconstruction and likelihood-based discrimination objectives used to train the agents and develop theories based on these objectives. Experiments are conducted using the MNIST dataset, and results are reported based on various metrics measuring semantic consistency and spatial meaningfulness.

**Strengths:**

- In the field of emergent communication, there is indeed a need to introduce concepts that capture the similarities with human language evolution and measures such as compositionally and efficiency have limitations in achieving this. In that regard, the topic of the paper is interesting and it useful to see semantic consistency being studied towards this goal.
- The paper is well-written, organized, and easy to read.
- The authors provide rigorous theoretical analysis and formal proofs regarding semantic consistency and spatial meaningfulness, which are robust and well-founded.
- The discussion of the approach's limitations is thorough and highly appreciated.

**Weaknesses:**

- While the introduction of a concept to analyze the likeness of emergent communication to the natural language (especially in terms of its meaningfulness or interpretability) is an interesting direction, there are several shortcomings of the current exposition with respect to its novelty. significance and clarity of the contributions.
- Semantic consistency definition seems highly contrived as it depends on distance between inputs which is only applicable to some modalities. The connection to human communication is weak and often far-fetched. The authors' claim that "inputs mapped to the same message should be semantically similar" is valid in some cases but does not universally apply to human language, making this motivation less useful.
- It is not clear why the difference observed in semantic consistency between reconstruction and discrimination task is surprising or an interesting result. The way it is discussed and formulated in this work, the implication on distance based similarity between inputs mapping to the same message (which are essentially latent representations of inputs learned with different objectives) seems to fall out from classical results on classification vs generative modeling.
- Along the similar lines, the spatial meaningfulness appears to be a fancy name for clustering of messages in the latent space. While the authors have overall weaker results for spatial meaningfulness, any difference in the structure of this latent space and resulting clusters also seem to fall out of classical deep learning concepts.
- While the authors already acknowledge various assumptions and simplifications required to present the analysis in this work as a part of limitations, the theorems on spatial meaningfulness requires even further stricter assumptions such a as simplicity and synchronization which limits its applicability.
- Related work: While the authors have covered a series of related literature, some important ones are missing. For instance, semantic meaning and other measures have been studied in several studies such as [1], [2] and [4]. Also, there is nice survey on this topic that is useful to position this work [6].
- A major drawback of this paper is that it considers a very simple and small setup and dataset for it to provide significant impact to the community where already emergent protocols are being studied at high scale [5] and complexity [3].
- The empirical results provide modest support for the theory, except for semantic consistency in the reconstruction task, as measured by message variance. However, the interpretation of message variance in this context is unclear.
- The omission of large language models (LLMs) as tools to measure the closeness of emergent protocols to human-like communication is a significant oversight. LLMs could serve as both evaluators and agents capable of generating more human-like communication protocols.


[1] Emergent Discrete Communication in Semantic Spaces, Tucker et. al. NeurIPS 2021

[2] Interpretable agent communication from scratch, Dessi et. al. NeurIPS 2021

[3] Emergent Communication for Rules Reasoning, Guo et. al. NeurIPS 2023

[4] Emergent communication for understanding human language evolution: what’s missing? Gelke et. al. ICLR 2022

[5] Emergent Communication at Scale, Chaabouni et. al. ICLR 2022

[6] Models of symbol emergence in communication: a conceptual review and a guide for avoiding local minima, Zubek et. al. 2023

**Questions:**

- Could the authors elaborate on the interpretability and usefulness of the explained variance?
- As mentioned earlier, most discussions and results are closely tied to the MNIST dataset. Could the authors discuss how their theory and measures would generalize to other datasets?
- How do you think such setups and measures would fare in the presence of LLMs? What are some examples where these measurement concepts would be useful, considering either the agents themselves can use LLMs or an LLM interpreter could make sense of the emergent protocols?
- Are the theoretical results tied specifically to two-agent games, or would they scale with the number of agents?
- As highlighted in the weaknesses, many results appear to be artifacts of the setup where the sender serves as an input encoder and the receiver acts either as a decoder (reconstruction) or predictor (classification), with messages representing the latent state space of inputs. Given this, could the authors explain the motivation and significance of their theoretical results? Is there some aspect of this setup and definition of semantic consistency that makes these results non-trivial?
- It is unclear if the study of spatial meaningfulness provides any useful insights, as both theoretical and empirical results seem incomplete and less rigorous. Could you elaborate on why no significant differences were found? One possible reason might be the reliance on the threshold value $\epsilon_0$. Have the authors studied the effect of changing this threshold?
- It may also be useful to explore how these measures relate to already established measures such as compositionality and efficiency. For instance, do semantically consistent messages lead to more efficient protocols or the emergence of a more compositional language?

**Limitations:**

The authors have discussed the limitations at length which is a plus point of the paper and remaining limitations have been described in the weakness section.

---

> ### Author Rebuttal · Authors · 2024-08-06
>
> Thank you for this insightful review. We are glad you have found the paper to be rigorous and well written. Following your feedback and others, we have prepared some major modifications to be added in the next revision.
>
> **Question 1**
>
> The explained variance relates to the average proximity of inputs with identical messages. The absence of explained variance (which we refer to as semantic inconsistency) means that inputs mapped to the same message are not closer to each other (in expectation) than two randomly selected inputs. This would indicate an inherently unintuitive communication protocol.
>
> **Question 2**
>
> Since our main contributions are theoretical, we used MNIST to generate proof-of-concept results. That said, given your question, we have conducted additional experiments on a more sophisticated dataset—a special version of the Shapes dataset [1] where each input is an image of an object with several sampled attributes (shape, color, position, rotation, and size). The results are appended to the general author rebuttal (and will be added to the paper in the next revision).
>
> The results show a significant gap in semantic consistency (lower message variance is preferable), on par with the previous evaluation on MNIST. Additionally, topographic similarity strongly favors the reconstruction setting (0.37, versus 0.08 for discrimination), confirming a positive relationship between semantic consistency (as measured by message variance) and compositionality (as measured by topsim). We also report correlations between every pair of measures.
>
> **Question 3**
>
> Interesting question. In general the field of EC as we know it did not catch up with the advance of LLMs. One might try to use LLMs for EC interpretation and decipherment, but this seems quite challenging. This idea is empirical in nature and not entirely in the spirit of our paper. In any case, we will add discussion of these possible connections in a revision.
>
> **Question 4**
>
> Our analysis only considers two-agent setups. The generalization of Lewis' games to multiple agents is not straightforward but could provide interesting insights.
>
> **Question 5**
>
> The sender-receiver EC setups are indeed built to resemble encoding and decoding, which is why we can consider messages as latent representations. What distinguishes EC setups from general representation learning is the discreteness of the communication channel, which is key to our analysis. The discreteness induces the many-to-one nature of the communication protocol, which is why the perspective of equivalence classes makes sense in this context. As for the significance of our (theoretical) results: a greatly desired goal in EC literature is the characterization of communication properties that separate human language from EC protocols. We discover, precisely define, and rigorously support two such properties, which can be used in future research. For example, our analysis led us to question the generative vs. discriminative division of objectives, in favor of a distance-based vs. probability-based distinction. This move is reflected in our choice to rebrand the probabilistic reconstruction variant (which is generative) as "global discrimination." These results are novel and non-trivial.
>
> **Question 6**
>
> Thank you for this insightful question. With regard to the empirical results, we admit that our proposed metric (cluster variance) for spatial meaningfulness does not yield interesting results. We hypothesize that the size of each cluster (16 messages) is too large, and perhaps a different partitioning method will provide more insight. This hypothesis will be added in the next revision. On the theoretical front, for the next revision we have improved our results by modifying the simplicity condition (ln 271) so that theorem 6.1 applies to any desired threshold (previous version applies to the minimal threshold $\varepsilon_M$). The interest behind the spatial meaningfulness definition is twofold: a) it allows an analysis that takes into account distances between messages, and b) the analysis does not assume an unrestricted hypothesis class, which is a major shortcoming of the theorems in section 5.
>
> **Question 7**
>
> There is an interesting relationship between our definitions and the most common compositionality measure, Topographic Similarity (topsim): topsim evaluates the correlation between distances in the input space and the corresponding distances in the message space. Notably, topsim considers the relationship between every pair of inputs, whereas our definitions only consider pairs that correspond to similar messages. The latter follows an intuitive asymmetry: inputs with similar messages are expected to be similar, but inputs with dissimilar messages do not have to be different. This paragraph will be added to the paper in the next revision. In addition, we performed additional experiments comparing message variance to existing compositionality measures from the literature. See the response to question 2.
>
> **The implication <...> seems to fall out from classical results on classification vs generative modeling.**
>
> An illustration of why this is not the case can be seen in the "global discrimination" game, which is generative but equivalent to a special case of the discrimination game (on the classification side).
>
> **Spatial meaningfulness appears to be a fancy name for clustering of messages in the latent space.**
>
> If anything, spatial meaningfulness describes the clustering of clusters! Recall that the message space M is given and constant; the sender only determines the mapping to messages. Each message corresponds to a set of inputs, and spatial meaningfulness requires those sets of inputs to be close (in the input space) when their message vectors are close.
>
> [1] Kuhnle, Alexander and Copestake, Ann. "ShapeWorld - A new test methodology for multimodal language understanding". arXiv preprint arXiv:1704.04517 (2017)

---

### Author Rebuttal · Authors · 2024-08-06

Dear reviewers,

We greatly appreciate the time and effort you have dedicated to evaluating our paper.

We have submitted individual responses to your reviews. To this message we have attached a pdf with new results on the Shapes dataset, along with visual illustrations of the trained agents' performance in each of the tasks.

---

### Decision · Program_Chairs · 2024-09-25

**Decision:**

Accept (poster)

**Comment:**

This paper investigates two goal-agnostic properties/prerequisites for meaningful emergent communication and supports the theoretical findings with results across discriminative & reconstructive tasks on the MNIST dataset, as proof-of-concept. The submission initially received mixed reviews from the four reviewers. All reviewers appreciated the well-written and interesting theoretical results that reveal fundamental properties of emergent communication. Beyond requesting clarifications, the reviewers raised concerns about inadequate experimental validation, missing references, and unclear mathematical formalization.

In the discussion phase, the authors addressed many of these concerns by adding experiments on a variation of the Shapes dataset, as suggested by reviewer o4TS, and providing detailed responses to questions from the reviewers. After the rebuttals, reviewers XigD and o4TS increased overall ratings, with o4TS indicating a move to a strong accept. Reviewer TFDu & cajw maintained their inclination to accept & reject, respectively.

The AC recommends acceptance and agrees with the majority of reviewers -- this work presents a novel theoretical understanding of emergent communication. While the committee has some residual concern regarding the insufficiency of empirical results, the paper’s current contributions hold significant value and are reasonably complete in theory, especially within the scope of NeurIPS, which recognizes new approaches. The authors are encouraged to improve the draft for the final version with a better discussion of spatial meaningfulness results (XigD, cajw), scalability and context of large-scale models (XigD, cajw), rigorous mathematical formalization (TFDu), and adding relevant citations & new experiments (o4TS) -- based on the constructive discussion between the authors and the reviewers.